# IDENTIFYING NONLINEAR DYNAMICAL SYSTEMS WITH MULTIPLE TIME SCALES AND LONG-RANGE DEPENDENCIES

**Dominik Schmidt**[1][*], **Georgia Koppe**[1,2][*], **Zahra Monfared**[1], **Max Beutelspacher**[1], **Daniel Durstewitz**[1,3][†]

## ABSTRACT

A main theoretical interest in biology and physics is to identify the nonlinear dynamical system (DS) that generated observed time series. Recurrent Neural Networks (RNNs) are, in principle, powerful enough to approximate any underlying DS, but in their vanilla form suffer from the exploding vs. vanishing gradients problem. Previous attempts to alleviate this problem resulted either in more complicated, mathematically less tractable RNN architectures, or strongly limited the dynamical expressiveness of the RNN. Here we address this issue by suggesting a simple regularization scheme for vanilla RNNs with ReLU activation which enables them to solve long-range dependency problems and express slow time scales, while retaining a simple mathematical structure which makes their DS properties partly analytically accessible. We prove two theorems that establish a tight connection between the regularized RNN dynamics and its gradients, illustrate on DS benchmarks that our regularization approach strongly eases the reconstruction of DS which harbor widely differing time scales, and show that our method is also en par with other long-range architectures like LSTMs on several tasks.

## 1 INTRODUCTION

Theories in the natural sciences are often formulated in terms of sets of stochastic differential or difference equations, i.e. as stochastic dynamical systems (DS). Such systems exhibit a range of common phenomena, like (limit) cycles, chaotic attractors, or specific bifurcations, which are the subject of nonlinear dynamical systems theory (DST; Strogatz (2015); Ott (2002)). A long-standing desire is to retrieve the generating dynamical equations directly from observed time series data (Kantz & Schreiber, 2004), and thus to 'automatize' the laborious process of scientific theory building to some degree. A variety of machine and deep learning methodologies toward this goal have been introduced in recent years (Chen et al., 2017; Champion et al., 2019; Ayed et al., 2019; Koppe et al., 2019; Hamilton et al., 2017; Razaghi & Paninski, 2019; Hernandez et al., 2020). Often these are based on sufficiently expressive series expansions for approximating the unknown system of generative equations, such as polynomial basis expansions (Brunton et al., 2016; Champion et al., 2019) or recurrent neural networks (RNNs) (Vlachas et al., 2018; Hernandez et al., 2020; Durstewitz, 2017; Koppe et al., 2019). Formally, RNNs are (usually discrete-time) nonlinear DS that are dynamically universal in the sense that they can approximate to arbitrary precision the flow field of any other DS on compact sets of the real space (Funahashi & Nakamura, 1993; Kimura & Nakano, 1998; Hanson & Raginsky, 2020). Hence, RNNs seem like a good choice for reconstructing – in this sense of *dynamically* equivalent behavior – the set of governing equations underlying real time series data.

However, RNNs in their vanilla form suffer from the 'vanishing or exploding gradients' problem (Hochreiter & Schmidhuber, 1997; Bengio et al., 1994): During training, error gradients tend to either exponentially explode or decay away across successive time steps, and hence vanilla RNNs face severe problems in capturing long time scales or long-range dependencies in the data. Specially designed RNN architectures equipped with gating mechanisms and linear memory cells have been proposed for mitigating this issue (Hochreiter & Schmidhuber, 1997; Cho et al., 2014). However, from a DST perspective, simpler models that can be more easily analyzed and interpreted in DS

---

[1]Department of Theoretical Neuroscience, [2]Clinic for Psychiatry and Psychotherapy, Central Institute of Mental Health, Medical Faculty Mannheim, Heidelberg University
[3]Faculty of Physics and Astronomy, Heidelberg University & Bernstein Center Computational Neuroscience
[*]These authors contributed equally
[†]Corresponding author: `daniel.durstewitz@zi-mannheim.de`

terms (Monfared & Durstewitz, 2020a;b), and for which more efficient inference algorithms exist that emphasize approximation of the true underlying DS (Koppe et al., 2019; Hernandez et al., 2020; Zhao & Park, 2020), would be preferable. More recent solutions to the vanishing vs. exploding gradient problem attempt to retain the simplicity of vanilla RNNs by initializing or constraining the recurrent weight matrix to be the identity (Le et al., 2015), orthogonal (Henaff et al., 2016; Helfrich et al., 2018) or unitary (Arjovsky et al., 2016). While merely initialization-based solutions, however, may be unstable and quickly dissolve during training, orthogonal or unitary constraints, on the other hand, are too restrictive for reconstructing DS, and more generally from a computational perspective as well (Kerg et al., 2019): For instance, neither chaotic behavior (that requires diverging directions) nor multi-stability, that is the coexistence of several distinct attractors, are possible.

Here we therefore suggest a different solution to the problem which takes inspiration from computational neuroscience: Supported by experimental evidence (Daie et al., 2015; Brody et al., 2003), line or plane attractors have been suggested as a dynamical mechanism for maintaining arbitrary information in working memory (Seung, 1996; Machens et al., 2005), a goal-related active form of short-term memory. A line or plane attractor is a continuous set of marginally stable fixed points to which the system's state converges from some neighborhood, while along the line itself there is neither con- nor divergence (Fig. 1**A**). Hence, a line attractor will perform a perfect integration of inputs and retain updated states indefinitely, while a slightly detuned line attractor will equip the system with arbitrarily slow time constants (Fig. 1**B**). This latter configuration has been suggested as a dynamical basis for neural interval timing (Durstewitz, 2003; 2004). The present idea is to exploit this dynamical setup for long short-term memory and arbitrary slow time scales by forcing part of the RNN's subspace toward a plane (line) attractor configuration through specifically designed regularization terms.

Specifically, our goal here is not so much to beat the state of the art on long short-term memory tasks, but rather to address the exploding vs. vanishing gradient problem within a simple, dynamically tractable RNN, optimized for DS reconstruction and interpretation. For this we build on piecewise-linear RNNs (PLRNNs) (Koppe et al., 2019; Monfared & Durstewitz, 2020b) which employ ReLU activation functions. PLRNNs have a simple mathematical structure (see eq. 1) which makes them dynamically *interpretable* in the sense that many geometric properties of the system's state space can in principle be computed analytically, including fixed points, cycles, and their stability (Suppl. 6.1.2; Koppe et al. (2019); Monfared & Durstewitz (2020a)), i.e. do not require numerical techniques (Sussillo & Barak, 2013). Moreover, PLRNNs constitute a type of piecewise linear (PWL) map for which many important bifurcations have been comparatively well characterized (Monfared & Durstewitz, 2020a; Avrutin et al., 2019). PLRNNs can furthermore be translated into equivalent continuous time ordinary differential equation (ODE) systems (Monfared & Durstewitz, 2020b) which comes with further advantages for analysis, e.g. continuous flow fields (Fig. 1**A**,**B**).

We retain the PLRNN's structural simplicity and analytical tractability while mitigating the exploding vs. vanishing gradient problem by adding special regularization terms for a subset of PLRNN units to the loss function. These terms are designed to push the system toward line attractor configurations, without strictly enforcing them, along some – but not all – directions in state space. We further establish a tight mathematical relationship between the PLRNN dynamics and the behavior of its gradients during training. Finally, we demonstrate that our approach outperforms LSTM and other, initialization-based, methods on a number of 'classical' machine learning benchmarks (Hochreiter & Schmidhuber, 1997). Much more importantly in the present DST context, we demonstrate that our new regularization-supported inference efficiently captures all relevant time scales when reconstructing challenging nonlinear DS with multiple short- and long-range phenomena.

## 2 RELATED WORK

*Dynamical systems reconstruction.* From a natural science perspective, the goal of reconstructing or identifying the underlying DS is substantially more ambitious than (and different from) building a system that 'merely' yields good ahead predictions: In DS identification we require that the inferred model can *freely reproduce* (when no longer guided by the data) the underlying attractor geometries and state space properties (see section 3.5, Fig. S2; Kantz & Schreiber (2004)).

Earlier work using RNNs for DS reconstruction (Roweis & Ghahramani, 2002; Yu et al., 2005) mainly focused on inferring the posterior over latent trajectories $\boldsymbol{Z} = \{\boldsymbol{z}_1, \ldots, \boldsymbol{z}_T\}$ given time series data $\boldsymbol{X} = \{\boldsymbol{x}_1, \ldots, \boldsymbol{x}_T\}$, $p(\boldsymbol{Z}|\boldsymbol{X})$, and on ahead predictions (Lu et al., 2017), as does much of

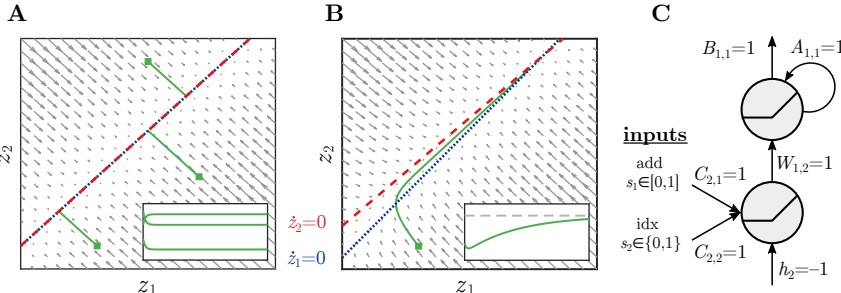

Figure 1: **A**)–**B**): Illustration of the state space of a 2-unit RNN with flow field (grey) and nullclines (set of points at which the flow of one of the variables vanishes, in blue and red). Insets: Time graphs of $z_1$ for $T = 30\,000$. **A**) Perfect line attractor. The flow converges to the line attractor, thus retaining states indefinitely in the absence of perturbations, as illustrated for 3 example trajectories (green). **B**) Slightly detuned line attractor. The system's state still converges toward the "attractor ghost", but then very slowly crawls up within the 'attractor tunnel' (green trajectory) until it hits the stable fixed point at the intersection of nullclines. Within the tunnel, flow velocity is smoothly regulated by the gap between nullclines, thus enabling arbitrary time constants. **C**) Simple 2-unit solution to the addition problem exploiting the line attractor properties of ReLUs. The output unit serves as a perfect integrator (see Suppl. 6.1.1 for complete parameters).

the recent work on variational inference of DS (Duncker et al., 2019; Zhao & Park, 2020; Hernandez et al., 2020). Although this enables insight into the dynamics along the empirically observed trajectories, both – posterior inference and good ahead predictions – do not per se guarantee that the inferred models can generate the underlying attractor geometries on their own (see Fig. S2, Koppe et al. (2019)). In contrast, if fully generative reconstruction of the underlying DS in this latter sense were achieved, formal analysis or simulation of the resulting RNN equations could provide a much deeper understanding of the dynamical mechanisms underlying empirical observations (Fig. 1 **C**).

Some approaches geared toward this latter goal of full DS reconstruction make specific structural assumptions about the form of the DS equations ('white box approach'; Meeds et al. (2019); Raissi (2018); Gorbach et al. (2017)), e.g. based on physical or biological domain knowledge, and focus on estimating the system's latent states and parameters, rather than approximating an unknown DS based on the observed time series information alone ('black box approach'). Others (Trischler & D'Eleuterio, 2016; Brunton et al., 2016; Champion et al., 2019) attempt to approximate the flow field, obtained e.g. by numerical differentiation, directly through basis expansions or neural networks. However, numerical derivatives are problematic for their high variance and other numerical issues (Raissi, 2018; Baydin et al., 2018; Chen et al., 2017). Another factor to consider is that in many biological systems like the brain the intrinsic dynamics are highly stochastic with many noise sources, like probabilistic synaptic release (Stevens, 2003). Models that do not explicitly account for dynamical process noise (Ayed et al., 2019; Champion et al., 2019; Rudy et al., 2019) are therefore less suited and more vulnerable to model misspecification. Finally, some fully probabilistic models for DS reconstruction based on GRU (Fraccaro et al., 2016), LSTM (Zheng et al., 2017; Vlachas et al., 2018), or radial basis function (Zhao & Park, 2020) networks, are not easily interpretable and amenable to DS analysis in the sense defined in sect. 3.3. Most importantly, none of these previous approaches consider the long-range dependency problem within more easily tractable RNNs for DS.

*Long-range dependency problems in RNNs*. Error gradients in vanilla RNNs tend to either explode or vanish due to the large product of derivative terms that results from recursive application of the chain rule over time steps (Hochreiter, 1991; Bengio et al., 1994; Hochreiter & Schmidhuber, 1997). To address this issue, RNNs with gated memory cells (Hochreiter & Schmidhuber, 1997; Cho et al., 2014) have been specifically designed, but their more complicated mathematical structure makes them less amenable to a systematic DS analysis. Even simple objects like fixed points of these systems have to be found by numerical techniques (Sussillo & Barak, 2013; Jordan et al., 2019). Thus, approaches which retain the simplicity of vanilla RNNs while solving the exploding vs. vanishing gradients problem would be desirable. Recently, Le et al. (2015) observed that initialization of the recurrent weight matrix $\boldsymbol{W}$ to the identity in ReLU-based RNNs may yield performance en par with LSTMs on standard machine learning benchmarks. Talathi & Vartak (2016) expanded on this idea by initializing the recurrence matrix such that its largest absolute eigenvalue is 1. Later work en-

forced orthogonal (Henaff et al., 2016; Helfrich et al., 2018; Jing et al., 2019) or unitary (Arjovsky et al., 2016) constraints on the recurrent weight matrix during training. While this appears to yield long-term memory performance sometimes superior to that of LSTMs (but see (Henaff et al., 2016)), these networks are limited in their computational power (Kerg et al., 2019). This may be a consequence of the fact that RNNs with orthogonal recurrence matrix are quite restricted in the range of dynamical phenomena they can produce, e.g. chaotic attractors are not possible since (locally) diverging eigen-directions are disabled.

Our approach therefore is to establish line/plane attractors only along some but not all directions in state space, and to only push the RNN toward these configurations but not strictly enforce them, such that convergence or (local) divergence of RNN dynamics is still possible. We furthermore implement these concepts through regularization terms in the loss functions, rather than through mere initialization. This way plane attractors are encouraged throughout training without fading away.

## 3 MODEL FORMULATION AND THEORETICAL ANALYSIS

### 3.1 BASIC MODEL FORMULATION

Assume we are given two multivariate time series $S = \{s_t\}$ and $X = \{x_t\}$, one we will denote as 'inputs' ($S$) and the other as 'outputs' ($X$). In the 'classical' (*supervised*) machine learning setting, we usually wish to map $S$ on $X$ through a RNN with latent state equation $z_t = F_\theta(z_{t-1}, s_t)$ and outputs $x_t \sim p_\lambda(x_t|z_t)$, as for instance in the 'addition problem' (Hochreiter & Schmidhuber, 1997). In DS reconstruction, in contrast, we usually have a dense time series $X$ from which we wish to infer (*unsupervised*) the underlying DS, where $S$ may provide an additional forcing function or sparse experimental inputs or perturbations. While our focus in this paper is on this latter task, DS reconstruction, we will demonstrate that our approach brings benefits in both these settings.

Here we consider for the latent model a PLRNN (Koppe et al., 2019) which takes the form

$$z_t = Az_{t-1} + W\phi(z_{t-1}) + Cs_t + h + \varepsilon_t, \ \ \varepsilon_t \sim \mathcal{N}(0, \Sigma), \tag{1}$$

where $z_t \in \mathbb{R}^{M \times 1}$ is the hidden state (column) vector of dimension $M$, $A \in \mathbb{R}^{M \times M}$ a *diagonal* and $W \in \mathbb{R}^{M \times M}$ an *off-diagonal* matrix, $s_t \in \mathbb{R}^{K \times 1}$ the external input of dimension $K$, $C \in \mathbb{R}^{M \times K}$ the input mapping, $h \in \mathbb{R}^{M \times 1}$ a bias, and $\varepsilon_t$ a Gaussian noise term with diagonal covariance matrix $\mathrm{diag}(\Sigma) \in \mathbb{R}_+^M$. The nonlinearity $\phi(z)$ is a ReLU, $\phi(z)_i = \max(0, z_i), i \in \{1, \dots, M\}$. This specific formulation represents a discrete-time version of firing rate (population) models as used in computational neuroscience (Song et al., 2016; Durstewitz, 2017; Engelken et al., 2020).

We will assume that the latent RNN states $z_t$ are coupled to the actual observations $x_t$ through a simple observation model of the form

$$x_t = Bg(z_t) + \eta_t, \ \eta_t \sim \mathcal{N}(0, \Gamma) \tag{2}$$

in the case of observations $x_t \in \mathbb{R}^{N \times 1}$, where $B \in \mathbb{R}^{N \times M}$ is a factor loading matrix, $g$ some (usually monotonic) nonlinear transfer function (e.g., ReLU), and $\mathrm{diag}(\Gamma) \in \mathbb{R}_+^N$ the diagonal covariance matrix of the Gaussian observation noise, or through a softmax function in case of categorical observations $x_{i,t} \in \{0, 1\}$ (see Suppl. 6.1.7 for details).

### 3.2 REGULARIZATION APPROACH

First note that by letting $A = I$, $W = 0$, and $h = 0$ in eq. 1, every point in $z$ space will be a *marginally stable* fixed point of the system, leading it to perform a perfect integration of external inputs as in parametric working memory (Machens et al., 2005; Brody et al., 2003).[1] This is similar in spirit to Le et al. (2015) who initialized RNN parameters such that it performs an identity mapping for $z_{i,t} \geq 0$. However, here 1) we use a neuroscientifically motivated network architecture (eq. 1) that enables the identity mapping across the variables' *entire support*, $z_{i,t} \in [-\infty, +\infty]$, which we conjecture will be of advantage for establishing long short-term memory properties, 2) we encourage

---

[1]Note that this very property of marginal stability required for input integration also makes the system sensitive to noise perturbations directly *on* the manifold attractor. Interestingly, this property has indeed been observed experimentally for real neural integrator systems (Major et al., 2004; Mizumori & Williams, 1993).

this mapping only for a subset $M_{\mathrm{reg}} \leq M$ of units (Fig. S1), leaving others free to perform arbitrary computations, and 3) we stabilize this configuration throughout training by introducing a specific $L_2$ regularization for parameters $\boldsymbol{A}$, $\boldsymbol{W}$, and $\boldsymbol{h}$ in eq. 1. When embedded into a larger, (locally) convergent system, we will call this configuration more generally a *manifold attractor*.

That way, we divide the units into two types, where the regularized units serve as a memory that tends to decay very slowly (depending on the size of the regularization term), while the remaining units maintain the flexibility to approximate any underlying DS, yet retaining the simplicity of the original PLRNN (eq. 1). Specifically, the following penalty is added to the loss function (Fig. S1):

$$\mathrm{L}_{\mathrm{reg}} = \tau_A \sum_{i=1}^{M_{\mathrm{reg}}} (A_{i,i} - 1)^2 + \tau_W \sum_{i=1}^{M_{\mathrm{reg}}} \sum_{\substack{j=1 \\ j \neq i}}^{M} W_{i,j}^2 + \tau_h \sum_{i=1}^{M_{\mathrm{reg}}} h_i^2 \tag{3}$$

(Recall from sect. 3.1 that $\boldsymbol{A}$ is a diagonal and $\boldsymbol{W}$ is an off-diagonal matrix.) While this formulation allows us to trade off, for instance, the tendency toward a manifold attractor ($\boldsymbol{A} \to \boldsymbol{I}$, $\boldsymbol{h} \to \boldsymbol{0}$) vs. the sensitivity to other units' inputs ($\boldsymbol{W} \to \boldsymbol{0}$), for all experiments performed here a common value, $\tau_A = \tau_W = \tau_h = \tau$, was assumed for the three regularization factors. We will refer to $(z_1 \ldots z_{M_{reg}})$ as the regularized ('memory') subsystem, and to $(z_{M_{reg}+1} \ldots z_M)$ as the non-regularized ('computational') subsystem. Note that in the limit $\tau \to \infty$ exact manifold attractors would be enforced.

### 3.3 THEORETICAL ANALYSIS

We will now establish a tight connection between the PLRNN dynamics and its error gradients. Similar ideas appeared in Chang et al. (2019), but these authors focused only on fixed point dynamics, while here we will consider the more general case including cycles of any order. First, note that by *interpretability* of model eq. 1 we mean that it is easily amenable to a rigorous DS analysis: As shown in Suppl. 6.1.2, we can explicitly determine all the system's fixed points and cycles and their stability. Moreover, as shown in Monfared & Durstewitz (2020b), we can – under certain conditions – transform the PLRNN into an equivalent continuous-time (ODE) piecewise-linear system, which brings further advantages for DS analysis.

Let us rewrite eq. 1 in the form

$$\boldsymbol{z}_t = F(\boldsymbol{z}_{t-1}) = (\boldsymbol{A} + \boldsymbol{W} \boldsymbol{D}_{\Omega(t-1)}) \boldsymbol{z}_{t-1} + \boldsymbol{h} := \boldsymbol{W}_{\Omega(t-1)} \boldsymbol{z}_{t-1} + \boldsymbol{h}, \tag{4}$$

where $\boldsymbol{D}_{\Omega(t-1)}$ is the diagonal matrix of outer derivatives of the ReLU function evaluated at $\boldsymbol{z}_{t-1}$ (see Suppl. 6.1.2), and we ignore external inputs and noise terms for now. Starting from some initial condition $\boldsymbol{z}_1$, we can recursively develop $\boldsymbol{z}_T$ as (see Suppl. 6.1.2 for more details):

$$\boldsymbol{z}_T = F^{T-1}(\boldsymbol{z}_1) = \prod_{i=1}^{T-1} \boldsymbol{W}_{\Omega(T-i)} \, \boldsymbol{z}_1 + \left[ \sum_{j=2}^{T-1} \prod_{i=1}^{j-1} \boldsymbol{W}_{\Omega(T-i)} + \boldsymbol{I} \right] \boldsymbol{h}. \tag{5}$$

Likewise, for some common loss function $\mathcal{L}(\boldsymbol{A}, \boldsymbol{W}, \boldsymbol{h}) = \sum_{t=2}^{T} \mathcal{L}_t$, we can recursively develop the derivatives w.r.t. weights $w_{mk}$ (and similar for components of $\boldsymbol{A}$ and $\boldsymbol{h}$) as

$$\frac{\partial \mathcal{L}}{\partial w_{mk}} = \sum_{t=2}^{T} \frac{\partial \mathcal{L}_t}{\partial \boldsymbol{z}_t} \frac{\partial \boldsymbol{z}_t}{\partial w_{mk}}, \quad \text{with } \frac{\partial \boldsymbol{z}_t}{\partial w_{mk}} = \mathbf{1}_{(m,k)} \boldsymbol{D}_{\Omega(t-1)} \boldsymbol{z}_{t-1} \tag{6}$$

$$+ \sum_{j=2}^{t-2} \left( \prod_{i=1}^{j-1} \boldsymbol{W}_{\Omega(t-i)} \right) \mathbf{1}_{(m,k)} \boldsymbol{D}_{\Omega(t-j)} \boldsymbol{z}_{t-j} + \prod_{i=1}^{t-2} \boldsymbol{W}_{\Omega(t-i)} \frac{\partial \boldsymbol{z}_2}{\partial w_{mk}},$$

where $\mathbf{1}_{(m,k)}$ is an $M \times M$ indicator matrix with a 1 for the $(m,k)$'th entry and 0 everywhere else. Observing that eqs. 5 and 6 contain similar product terms which determine the system's long-term behavior, our first theorem links the PLRNN dynamics to its total error gradients:

**Theorem 1.** *Consider a PLRNN given by eq. 4, and assume that it converges to a stable fixed point, say $\boldsymbol{z}_{t*1} := \boldsymbol{z}^{*1}$, or a k-cycle ($k > 1$) with the periodic points $\{\boldsymbol{z}_{t*k}, \boldsymbol{z}_{t*k-1}, \cdots, \boldsymbol{z}_{t*k-(k-1)}\}$, for $T \to \infty$. Suppose that, for $k \geq 1$ and $i \in \{0, 1, \cdots, k-1\}$, $\sigma_{max}(\boldsymbol{W}_{\Omega(t*k-i)}) = \|\boldsymbol{W}_{\Omega(t*k-i)}\| < 1$, where $\boldsymbol{W}_{\Omega(t*k-i)}$ denotes the Jacobian of the system at $\boldsymbol{z}_{t*k-i}$ and $\sigma_{max}$ indicates the largest*

*singular value of a matrix. Then, the 2-norms of the tensors collecting all derivatives,* $\left\|\frac{\partial \boldsymbol{z}_T}{\partial \boldsymbol{W}}\right\|_2$, $\left\|\frac{\partial \boldsymbol{z}_T}{\partial \boldsymbol{A}}\right\|_2$, $\left\|\frac{\partial \boldsymbol{z}_T}{\partial \boldsymbol{h}}\right\|_2$, *will be bounded from above, i.e. will not diverge for* $T \to \infty$.

*Proof.* See Suppl. sect. 6.1 (subsection 6.1.3). □

While Theorem 1 is a general statement about PLRNN dynamics and total gradients, our next theorem more specifically provides conditions under which Jacobians linking temporally distant states $\boldsymbol{z}_T$ and $\boldsymbol{z}_t$, $T \gg t$, will neither vanish nor explode in the regularized PLRNN:

**Theorem 2.** *Assume a PLRNN with matrix* $\boldsymbol{A} + \boldsymbol{W}$ *partitioned as in Fig. S1, i.e. with the first* $M_{reg}$ *rows corresponding to those of an* $M \times M$ *identity matrix. Suppose that the non-regularized subsystem* $(z_{M_{reg}+1} \ldots z_M)$, *if considered in isolation, satisfies Theorem 1, i.e. converges to a k-cycle with* $k \geq 1$. *Then, for the full system* $(z_1 \ldots z_M)$, *the 2-norm of the Jacobians connecting temporally distal states* $\boldsymbol{z}_T$ *and* $\boldsymbol{z}_t$ *will be bounded from above and below for all* $T > t$, *i.e.* $\infty > \rho_{up} \geq \left\|\frac{\partial \boldsymbol{z}_T}{\partial \boldsymbol{z}_t}\right\|_2 = \left\|\prod_{t < k \leq T} \boldsymbol{W}_{\Omega(k)}\right\|_2 \geq \rho_{low} > 0$. *In particular, for state variables* $z_{iT}$ *and* $z_{jt}$ *such that* $i \in \{M_{reg} + 1, \cdots, M\}$ *and* $j \in \{1, \cdots, M_{reg}\}$, *i.e. that connect states from the 'memory' to those of the 'computational' subsystem, one also has* $\infty > \lambda_{up} \geq \left|\frac{\partial z_{iT}}{\partial z_{jt}}\right| \geq \lambda_{low} > 0$ *as* $T - t \to \infty$, *i.e. these derivatives will never vanish nor explode.*

*Proof.* See Suppl. sect. 6.1 (subsection 6.1.4). □

The bounds $\rho_{up}, \rho_{low}, \lambda_{up}, \lambda_{low}$, are given in Suppl. sect. 6.1.4. We remark that when the regularization conditions are not exactly met, i.e. when parameters $\boldsymbol{A}$ and $\boldsymbol{W}$ slightly deviate from those in Fig. S1, memory (and gradients) may ultimately dissipate, but only very slowly, as actually required for temporal processes with very slow yet not infinite time constants (Fig. 1**B**).

## 3.4 TRAINING PROCEDURES

For the (supervised) machine learning problems, all networks were trained by stochastic gradient descent (SGD) to minimize the squared-error loss between estimated and actual outputs for the addition and multiplication problems, and the cross entropy loss for sequential MNIST (see Suppl. 6.1.7). Adam (Kingma & Ba, 2014) from PyTorch package (Paszke et al., 2017) was used as the optimizer, with a learning rate of 0.001, gradient clip parameter of 10, and batch size of 500. SGD was stopped after 100 epochs and the fit with the lowest loss across all epochs was taken, except for LSTM which was allowed to run for up to 200 epochs as it took longer to converge (Fig. S10). For comparability, the PLRNN latent state dynamics eq. 1 was assumed to be deterministic in this setting (i.e., $\boldsymbol{\Sigma} = \boldsymbol{0}$), $g(\boldsymbol{z}_t) = \boldsymbol{z}_t$ and $\boldsymbol{\Gamma} = \boldsymbol{I}_N$ in eq. 2. For the regularized PLRNN (rPLRNN), penalty eq. 3 was added to the loss function. For the (unsupervised) DS reconstruction problems, the fully probabilistic, generative RNN eq. 1 was considered. Together with eq. 2 (where we take $g(\boldsymbol{z}_t) = \phi(\boldsymbol{z}_t)$) this gives the typical form of a nonlinear state space model (Durbin & Koopman, 2012) with observation and process noise, and an Expectation-Maximization (EM) algorithm that efficiently exploits the model's piecewise linear structure (Durstewitz, 2017; Koppe et al., 2019) was used to solve for the parameters by maximum likelihood. Details are given in Suppl. 6.1.5. All code used here will be made openly available at https://github.com/DurstewitzLab/reg-PLRNN.

## 3.5 PERFORMANCE MEASURES

For the machine learning benchmarks we employed the same criteria as used for optimization (MSE or cross-entropy, Suppl. 6.1.7) as performance metrics, evaluated across left-out test sets. In addition, we report the relative frequency $P_{\text{correct}}$ of correctly predicted trials across the test set (see Suppl. 6.1.7 for details). For DS reconstruction problems, it is not sufficient or even sensible to judge a method's ability to infer the underlying DS purely based on some form of (ahead-)prediction error like the MSE defined on the time series itself (Ch.12 in Kantz & Schreiber (2004)). Rather, we require that the inferred model can freely reproduce (when no longer guided by the data) the underlying attractor geometries and state space properties. This is not automatically guaranteed for a model that yields agreeable ahead predictions on a time series (Fig. S2**A**; cf. Koppe et al. (2019); Wood (2010)). We therefore followed Koppe et al. (2019) and used the Kullback-Leibler divergence between true and reproduced probability distributions across states in state space to quantify how well an inferred PLRNN captured the underlying dynamics, thus assessing the agreement in attractor geometries (cf. Takens (1981); Sauer et al. (1991)) (see Suppl. 6.1.6 for more details).

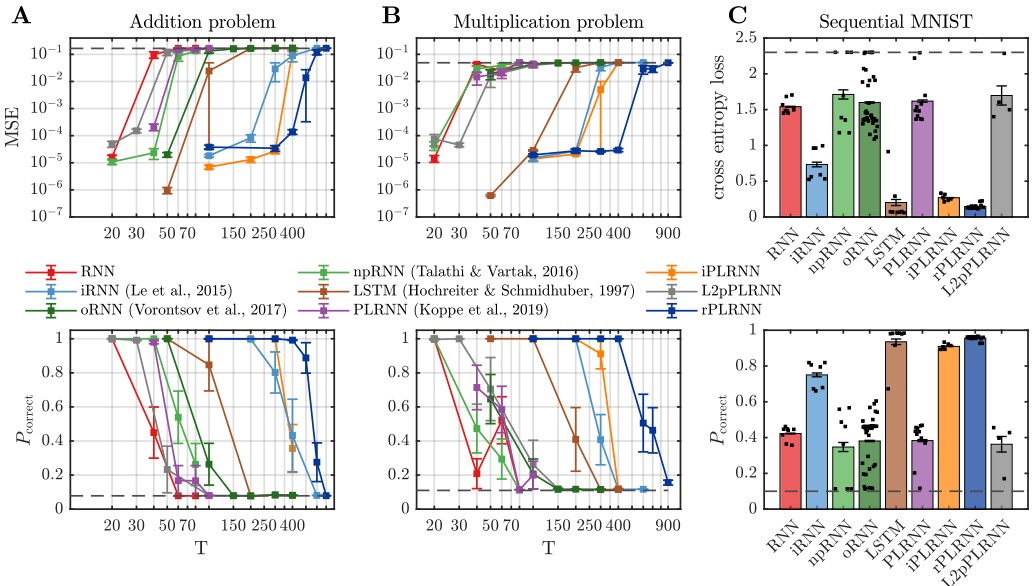

Figure 2: Comparison of rPLRNN ($\tau = 5$, $\frac{M_{\text{reg}}}{M} = 0.5$, cf. Fig. S3) to other methods for **A**) addition problem, **B**) multiplication problem and **C**) sequential MNIST. Top row gives loss as a function of time series length $T$ (error bars = SEM, $n \geq 5$), bottom row shows relative frequency of correct trials. Note that better performance (lower values in top row, higher values in bottom row) is reflected in a more rightward shift of curves. Dashed lines indicate chance level, black dots in **C** indicate individual repetitions.

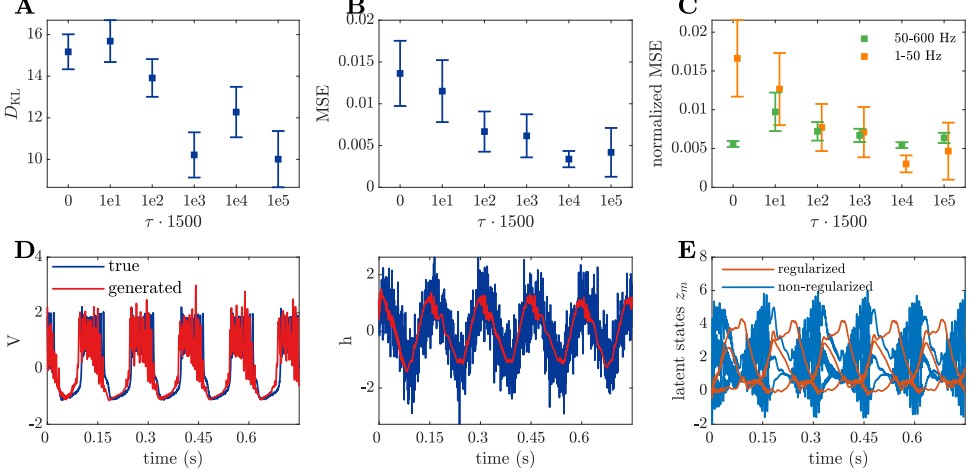

Figure 3: Reconstruction of a 2-time scale DS in limit cycle regime. **A**) KL divergence ($D_{\text{KL}}$) between true and generated state space distributions. Globally diverging system estimates were removed. **B**) Average MSE between power spectra of true and reconstructed DS and **C**) split according to low ($\leq 50$ Hz) and high ($> 50$ Hz) frequency components. Error bars = SEM ($n = 33$). **D**) Example of (best) generated time series (red=reconstruction with $\tau = \frac{2}{3}$). See Fig. S5A for variable $n$. **E**) Dynamics of regularized and non-regularized latent states for the example in **D**.

## 4 NUMERICAL EXPERIMENTS

### 4.1 MACHINE LEARNING BENCHMARKS

Although not our prime interest here, we first examined how the rPLRNN would fare on supervised machine learning benchmarks where inputs ($S$) are to be mapped onto target outputs ($X$) across long

time spans (i.e., requiring long short-term maintenance of information), namely the *addition* and *multiplication problems* (Talathi & Vartak, 2016; Hochreiter & Schmidhuber, 1997), and *sequential MNIST* (LeCun et al., 2010). Details of these experimental setups are in Suppl. 6.1.7. Performance of the rPLRNN (eq. 1, eq. 3) on all 3 benchmarks was compared to several other models summarized in Suppl. Table 1. To achieve a meaningful comparison, all models have the same number $M = 40$ (based on Fig. S3) of hidden states (which gives LSTMs overall about 4 times as many trainable parameters). On all three problems the rPLRNN outperforms all other tested methods, including LSTM, iRNN (RNN initialized by the identity matrix as in Le et al. (2015)), and a version of the orthogonal RNN (oRNN; Vorontsov et al. (2017)) (similar results were obtained for other settings of $M$ and batch size). LSTM performs even worse than iRNN and iPLRNN (PLRNN initialized with the identity as the iRNN), although it had 4 times as many parameters and was given twice as many epochs (and thus opportunities) for training, as it also took longer to converge (Fig. S10). In addition, the iPLRNN tends to perform slightly better than the iRNN on all three problems, suggesting that the specific structure eq. 1 of the PLRNN that allows for a manifold attractor across the variables' full range may be advantageous to begin with, while the regularization further improves performance.

## 4.2 NUMERICAL EXPERIMENTS ON DYNAMICAL SYSTEMS WITH DIFFERENT TIME SCALES

While it is encouraging that the rPLRNN may perform even better than several previous approaches to the vanishing vs. exploding gradients problem, our major goal here was to examine whether our regularization scheme would help with the (unsupervised) identification of DS that harbor widely different time scales. To test this, we used a biophysical, bursting cortical neuron model with one voltage ($V$) and two conductance recovery variables (see Durstewitz (2009)), one slow ($h$) and one fast ($n$; Suppl. 6.1.8). Reproduction of this DS is challenging since it produces very fast spikes on top of a slow nonlinear oscillation (Fig. 3**D**). Only short time series (as in scientific data) of length $T = 1500$ from this model were provided for training. rPLRNNs with $M = \{8 \ldots 18\}$ states were trained, with the regularization factor varied within $\tau \in \{0, 10^1, 10^2, 10^3, 10^4, 10^5\}/T$. Note that for $\tau = 0$ (no regularization), the approach reduces to the standard PLRNN (Koppe et al., 2019).

Fig. 3**A** confirms our intuition that stronger regularization leads to better DS reconstruction as assessed by the KL divergence between true and generated state distributions (similar results were obtained with ahead-prediction errors as a metric, Fig. S4**A**), accompanied by a likewise decrease in the MSE between the power spectra of true (suppl. eq. 55) and generated (rPLRNN) voltage traces (Fig. 3**B**). Fig. 3**D** gives an example of voltage traces ($V$) and the slower of the two gating variables ($h$; see Fig. S5**A** for variable $n$) freely simulated (i.e., sampled) from the autonomously running rPLRNN. This illustrates that our model is in principle capable of capturing both the stiff spike dynamics and the slower oscillations in the second gating variable at the same time. Fig. 3**C** provides more insight into how the regularization worked: While the high frequency components ($> 50$ Hz) related to the repetitive spiking activity hardly benefited from increasing $\tau$, there was a strong reduction in the MSE computed on the power spectrum for the lower frequency range ($\leq 50$ Hz), suggesting that increased regularization helps to map slowly evolving components of the dynamics. This result is more general as shown in Fig. S6 for another DS example. In contrast, an orthogonality (Vorontsov et al., 2017) or plain L2 constraint on weight matrices did not help at all on this problem (Fig. S4**B**).

Further insight into the dynamical mechanisms by which the rPLRNN solves the problem can be obtained by examining the latent dynamics: As shown in Fig. 3**E** (see also Fig. S5), regularized states indeed help to map the slow components of the dynamics, while non-regularized states focus on the fast spikes. These observations further corroborate the findings in Fig. 3**C** and Fig. S6**C**.

## 4.3 REGULARIZATION PROPERTIES AND MANIFOLD ATTRACTORS

In Figs. 2 and 3 we demonstrated that the rPLRNN is able to solve problems and reconstruct dynamics that involve long-range dependencies. Figs. 3**A**,**B** furthermore directly confirm that solutions improve with stronger regularization, while Figs. 3**C**,**E** give insight into the mechanism by which the regularization works. To further verify empirically that our specific form of regularization, eq. 3, is important, Fig. 2 also shows results for a PLRNN with standard L2 norm on a fraction of $M_{reg}/M = 0.5$ states (L2pPLRNN). Fig. S7 provides additional results for PLRNNs with L2 norm on all weights and for vanilla L2-regularized RNNs. All these systems fell far behind the performance of the rPLRNN on all tasks tested. Moreover, Fig. 4 reveals that the specific regularization

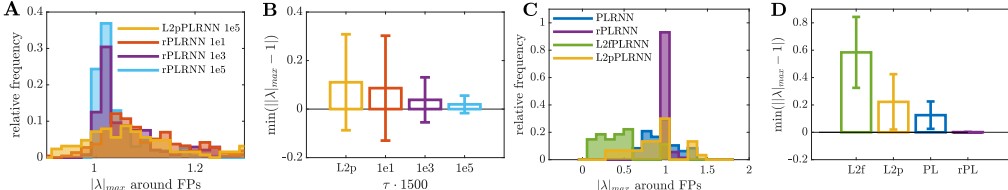

Figure 4: **A**) Distribution of maximum absolute eigenvalues $\lambda$ of Jacobians around fixed points for rPLRNN for different $\tau$ and L2PLRNN trained on bursting neuron DS. **B**) Absolute deviations of max. $|\lambda|$ from 1 (using for each system the one eigenvalue with smallest deviation). **C**) Same as **A** for addition problem for rPLRNN ($\tau = 5$) vs. standard, fully L2- (L2f), and partially L2 (L2p)-regularized PLRNN. **D**) Same as **B** for the models from **C**. Error bars = stdv. See also Fig. S8.

proposed indeed encourages manifold attractors, and that this is not achieved by a standard L2 regularization: In contrast to L2PLRNN, as the regularization factor $\tau$ is increased, more and more of the maximum absolute eigenvalues around the system's fixed points (computed according to eq. 8, sect. 6.1.2) cluster on or near 1, indicating directions of marginal stability in state space. Also, the deviations from 1 become smaller for strongly regularized PLRNNs (Fig. 4**B,D**), indicating a higher precision in attractor tuning. Fig. S9 in addition confirms that rPLRNN parameters are increasingly driven toward values that would support manifold attractors with stronger regularization. Fig. 3**E** furthermore suggests that both regularized and non-regularized states are utilized to map the full dynamics. But how should the ratio $M_{\mathrm{reg}}/M$ be chosen in practice? While for the problems here this meta-parameter was determined through 'classical' grid-search and cross-validation, Figs. S3 **C** – **E** suggest that the precise setting of $M_{\mathrm{reg}}/M$ is actually not overly important: Nearly optimal performance is achieved for a broader range $M_{\mathrm{reg}}/M \in [0.3, 0.6]$ on all problems tested. Hence, in practice, setting $M_{\mathrm{reg}}/M = 0.5$ should mostly work fine.

## 5 CONCLUSIONS

In this work we introduced a simple solution to the long short-term memory problem in RNNs that retains the simplicity and tractability of PLRNNs, yet does not curtail their universal computational capabilities (Koiran et al., 1994; Siegelmann & Sontag, 1995) and their ability to approximate arbitrary DS (Funahashi & Nakamura, 1993; Kimura & Nakano, 1998; Trischler & D'Eleuterio, 2016). We achieved this by adding regularization terms to the loss function that encourage the system to form a 'memory subspace' (Seung, 1996; Durstewitz, 2003) which would store arbitrary values for, if unperturbed, arbitrarily long periods. At the same time we did not rigorously enforce this constraint, which allowed the system to capture slow time scales by slightly departing from a perfect manifold attractor. In neuroscience, this has been discussed as a dynamical mechanism for regulating the speed of flow in DS and learning of arbitrary time constants not naturally included qua RNN design (Durstewitz, 2003; 2004) (Fig. 1**B**). While other RNN architectures, including vanilla RNNs, can, in principle, also develop line attractors to solve specific tasks (Maheswaranathan et al., 2019), they are generally much harder to train to achieve this and may exhibit less precise attractor tuning (cf. Fig. 4), which is needed to bridge long time scales (Durstewitz, 2003). Moreover, part of the PLRNN's latent space was not regularized at all, leaving the system enough degrees of freedom for realizing arbitrary computations or dynamics (see also Fig. S11 for a chaotic example). We showed that the rPLRNN is en par with or outperforms initialization-based approaches, orthogonal RNNs, and LSTMs on a number of classical benchmarks. More importantly, however, the regularization strongly facilitates the identification of challenging DS with widely different time scales in PLRNN-based algorithms for DS reconstruction. Similar regularization schemes as proposed here (eq. 3) may, in principle, also be designed for other architectures, but the convenient mathematical form of the PLRNN makes their implementation particularly powerful and straightforward.

### ACKNOWLEDGEMENTS

This work was funded by grants from the German Research Foundation (DFG) to DD (Du 354/10-1, Du 354/8-2 within SPP 1665) and to GK (TRR265: A06 & B08), and under Germany's Excellence Strategy – EXC-2181 – 390900948 ('Structures').

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

## 6 APPENDIX

### 6.1 SUPPLEMENTARY TEXT

#### 6.1.1 *Simple exact PLRNN solution for addition problem*

The exact PLRNN parameter settings (cf. eq. 1, eq. 2) for solving the addition problem with 2 units (cf. Fig. 1**C**) are as follows:

$$\boldsymbol{A} = \begin{pmatrix} 1 & 0 \\ 0 & 0 \end{pmatrix}, \boldsymbol{W} = \begin{pmatrix} 0 & 1 \\ 0 & 0 \end{pmatrix}, \boldsymbol{h} = \begin{pmatrix} 0 \\ -1 \end{pmatrix}, \boldsymbol{C} = \begin{pmatrix} 0 & 0 \\ 1 & 1 \end{pmatrix}, \boldsymbol{B} = \begin{pmatrix} 1 & 0 \end{pmatrix} \tag{7}$$

#### 6.1.2 *Computation of fixed points and cycles in PLRNN*

Consider the PLRNN in the form of eq. 4. For clarity, let us define $\boldsymbol{d}_{\Omega(t)} := (d_1, d_2, \cdots, d_M)$ as an indicator vector with $d_m(z_{m,t}) := d_m = 1$ for all states $z_{m,t} > 0$ and zeros otherwise, and $\boldsymbol{D}_{\Omega(t)} := \mathrm{diag}(\boldsymbol{d}_{\Omega(t)})$ as the diagonal matrix formed from this vector. Note that there are at most $2^M$ distinct matrices $\boldsymbol{W}_{\Omega(t)}$ as defined in eq. 4, depending on the sign of the components of $\boldsymbol{z}_t$.

If $\boldsymbol{h} = \boldsymbol{0}$ and $\boldsymbol{W}_{\Omega(t)}$ is the identity matrix, then the map $F$ becomes the identity map and so every point $\boldsymbol{z}$ will be a fixed point of $F$. Otherwise, the fixed points of $F$ can be found solving the equation $F(\boldsymbol{z}^{*1}) = \boldsymbol{z}^{*1}$ as

$$\boldsymbol{z}^{*1} = (\boldsymbol{I} - \boldsymbol{W}_{\Omega(t^{*1})})^{-1} \boldsymbol{h} = \boldsymbol{H}^{*1} \boldsymbol{h}, \tag{8}$$

where $\boldsymbol{z}^{*1} = \boldsymbol{z}_{t^{*1}} = \boldsymbol{z}_{t^{*1}-1}$, if $\det(\boldsymbol{I} - \boldsymbol{W}_{\Omega(t^{*1})}) = P_{\boldsymbol{W}_{\Omega(t^{*1})}}(1) \neq 0$, i.e. $\boldsymbol{W}_{\Omega(t^{*1})}$ has no eigenvalue equal to 1. Stability and type of fixed points (node, saddle, spiral) can then be determined from the eigenvalues of the Jacobian $\boldsymbol{A} + \boldsymbol{W}\boldsymbol{D}_{\Omega(t^{*1})} = \boldsymbol{W}_{\Omega(t^{*1})}$ (Strogatz (2015)).

For $k > 1$, solving $F^k(\boldsymbol{z}^{*k}) = \boldsymbol{z}^{*k}$, one can obtain a $k$-cycle of the map $F$ with the periodic points $\{\boldsymbol{z}^{*k}, F(\boldsymbol{z}^{*k}), F^2(\boldsymbol{z}^{*k}), \cdots, F^{k-1}(\boldsymbol{z}^{*k})\}$. For this, we first compute $F^k$ as follows:

$$\boldsymbol{z}_t = F(\boldsymbol{z}_{t-1}) = \boldsymbol{W}_{\Omega(t-1)} \boldsymbol{z}_{t-1} + \boldsymbol{h},$$

$$\boldsymbol{z}_{t+1} = F^2(\boldsymbol{z}_{t-1}) = F(\boldsymbol{z}_t) = \boldsymbol{W}_{\Omega(t)} \boldsymbol{W}_{\Omega(t-1)} \boldsymbol{z}_{t-1} + \big(\boldsymbol{W}_{\Omega(t)} + \boldsymbol{I}\big)\boldsymbol{h},$$

$$\boldsymbol{z}_{t+2} = F^3(\boldsymbol{z}_{t-1}) = F(\boldsymbol{z}_{t+1}) = \boldsymbol{W}_{\Omega(t+1)} \boldsymbol{W}_{\Omega(t)} \boldsymbol{W}_{\Omega(t-1)} \boldsymbol{z}_{t-1}$$
$$+ \big(\boldsymbol{W}_{\Omega(t+1)} \boldsymbol{W}_{\Omega(t)} + \boldsymbol{W}_{\Omega(t+1)} + \boldsymbol{I}\big)\boldsymbol{h},$$

$$\vdots$$

$$\boldsymbol{z}_{t+(k-1)} = F^k(\boldsymbol{z}_{t-1}) = \prod_{i=2}^{k+1} \boldsymbol{W}_{\Omega(t+(k-i))} \boldsymbol{z}_{t-1} + \left[ \sum_{j=2}^{k} \prod_{i=2}^{k-j+2} \boldsymbol{W}_{\Omega(t+(k-i))} + \boldsymbol{I} \right] \boldsymbol{h}, \tag{9}$$

in which $\prod_{i=2}^{k+1} \boldsymbol{W}_{\Omega(t+(k-i))} = \boldsymbol{W}_{\Omega(t+(k-2))} \boldsymbol{W}_{\Omega(t+(k-3))} \cdots \boldsymbol{W}_{\Omega(t-1)}$.
Assuming $t + (k-1) := t^{*k}$, then the $k$-cycle is given by the fixed point of the $k$-times iterated map $F^k$ as

$$\boldsymbol{z}^{*k} = \left( \boldsymbol{I} - \prod_{i=1}^{k} \boldsymbol{W}_{\Omega(t^{*k}-i)} \right)^{-1} \left[ \sum_{j=2}^{k} \prod_{i=1}^{k-j+1} \boldsymbol{W}_{\Omega(t^{*k}-i)} + \boldsymbol{I} \right] \boldsymbol{h} = \boldsymbol{H}^{*k} \boldsymbol{h}, \tag{10}$$

where $\boldsymbol{z}^{*k} = \boldsymbol{z}_{t^{*k}} = \boldsymbol{z}_{t^{*k}-k}$, provided that $\boldsymbol{I} - \prod_{i=1}^{k} \boldsymbol{W}_{\Omega(t^{*k}-i)}$ is invertible. That is $\det\left( \boldsymbol{I} - \prod_{i=1}^{k} \boldsymbol{W}_{\Omega(t^{*k}-i)} \right) = P_{\prod_{i=1}^{k} \boldsymbol{W}_{\Omega(t^{*k}-i)}}(1) \neq 0$ and $\prod_{i=1}^{k} \boldsymbol{W}_{\Omega(t^{*k}-i)} := \boldsymbol{W}_{\Omega^{*k}}$ has no eigenvalue equal to 1. As for the fixed points, we can determine stability of the $k$-cycle from the eigenvalues of the Jacobians $\prod_{i=1}^{k} \boldsymbol{W}_{\Omega(t^{*k}-i)}$.

It may also be helpful to spell out the recursions in eq. 5 and eq. 6 in section 3.3 in a bit more detail. Analogously to the derivations above, for $t = 1, 2, \ldots, T$ we can recursively compute $z_2, z_3, \ldots, z_T$ ($T \in \mathbb{N}$) as

$$z_2 = F(z_1) = W_{\Omega(1)} z_1 + h,$$

$$z_3 = F^2(z_1) = F(z_2) = W_{\Omega(2)} W_{\Omega(1)} z_1 + (W_{\Omega(2)} + I)h,$$

$$\vdots$$

$$z_T = F^{T-1}(z_1) = F(z_{T-1}) = W_{\Omega(T-1)} W_{\Omega(T-2)} \cdots W_{\Omega(1)} z_1$$
$$+ \big(W_{\Omega(T-1)} W_{\Omega(T-2)} \cdots W_{\Omega(2)}$$
$$+ W_{\Omega(T-1)} W_{\Omega(T-2)} \cdots W_{\Omega(3)} + \cdots + W_{\Omega(T-1)} + I\big)h$$

$$= \prod_{i=1}^{T-1} W_{\Omega(T-i)} z_1 + \bigg[ \sum_{j=1}^{T-2} \prod_{i=1}^{T-j-1} W_{\Omega(T-i)} + I \bigg] h$$

$$= \prod_{i=1}^{T-1} W_{\Omega(T-i)} z_1 + \bigg[ \sum_{j=2}^{T-1} \prod_{i=1}^{j-1} W_{\Omega(T-i)} + I \bigg] h. \tag{11}$$

Likewise, we can write out the derivatives eq. 6 more explicitly as

$$\frac{\partial z_t}{\partial w_{mk}} = \frac{\partial F(z_{t-1})}{\partial w_{mk}} = \mathbf{1}_{(m,k)} D_{\Omega(t-1)} z_{t-1} + \big(A + W D_{\Omega(t-1)}\big) \frac{\partial z_{t-1}}{\partial w_{mk}}$$

$$= \mathbf{1}_{(m,k)} D_{\Omega(t-1)} z_{t-1} + \big(A + W D_{\Omega(t-1)}\big) \mathbf{1}_{(m,k)} D_{\Omega(t-2)} z_{t-2}$$

$$+ \big(A + W D_{\Omega(t-1)}\big)\big(A + W D_{\Omega(t-2)}\big) \frac{\partial z_{t-2}}{\partial w_{mk}}$$

$$= \mathbf{1}_{(m,k)} D_{\Omega(t-1)} z_{t-1} + \big(A + W D_{\Omega(t-1)}\big) \mathbf{1}_{(m,k)} D_{\Omega(t-2)} z_{t-2}$$

$$+ \big(A + W D_{\Omega(t-1)}\big)\big(A + W D_{\Omega(t-2)}\big) \mathbf{1}_{(m,k)} D_{\Omega(t-3)} z_{t-3}$$

$$+ \big(A + W D_{\Omega(t-1)}\big)\big(A + W D_{\Omega(t-2)}\big)\big(A + W D_{\Omega(t-3)}\big) \frac{\partial z_{t-3}}{\partial w_{mk}}$$

$$= \cdots$$

$$= \mathbf{1}_{(m,k)} D_{\Omega(t-1)} z_{t-1} + \sum_{j=2}^{t-2} \bigg( \prod_{i=1}^{j-1} W_{\Omega(t-i)} \bigg) \mathbf{1}_{(m,k)} D_{\Omega(t-j)} z_{t-j}$$

$$+ \prod_{i=1}^{t-2} W_{\Omega(t-i)} \frac{\partial z_2}{\partial w_{mk}} \tag{12}$$

where $\frac{\partial z_2}{\partial w_{mk}} = (\frac{\partial z_{1,2}}{\partial w_{mk}} \cdots \frac{\partial z_{M,2}}{\partial w_{mk}})$ with $\frac{\partial z_{l,2}}{\partial w_{mk}} = 0 \,\forall l \neq m$ and $\frac{\partial z_{m,2}}{\partial w_{mk}} = d_k z_{k,1}$. The derivatives w.r.t. the elements of $A$ and $h$ can be expanded in a similar way, only that the terms $D_{\Omega(t)} z_t$ on the last line of eq. 12 need to be replaced by just $z_t$ for $\frac{\partial z_t}{\partial a_{mm}}$, and by just a vector of 1's for $\frac{\partial z_t}{\partial h_m}$ (also, in these cases, the indicator matrix will be the diagonal matrix $\mathbf{1}_{(m,m)}$).

### 6.1.3 *Proof of Theorem 1*

To state the proof, let us rewrite the derivatives of the loss function $\mathcal{L}(W, A, h) = \sum_{t=1}^{T} \mathcal{L}_t$ in the following tensor form:

$$\frac{\partial \mathcal{L}}{\partial W} = \sum_{t=1}^{T} \frac{\partial \mathcal{L}_t}{\partial W}, \quad \text{where} \quad \frac{\partial \mathcal{L}_t}{\partial W} = \frac{\partial \mathcal{L}_t}{\partial z_t} \frac{\partial z_t}{\partial W}, \tag{13}$$

for which the 3D tensor

$$\frac{\partial \boldsymbol{z}_t}{\partial \boldsymbol{W}} = \begin{pmatrix} \frac{\partial z_{1,t}}{\partial \boldsymbol{W}} \\ \frac{\partial z_{2,t}}{\partial \boldsymbol{W}} \\ \vdots \\ \frac{\partial z_{M,t}}{\partial \boldsymbol{W}} \end{pmatrix} \tag{14}$$

of dimension $M \times M \times M$, consists of all the gradient matrices

$$\frac{\partial z_{i,t}}{\partial \boldsymbol{W}} = \begin{pmatrix} \frac{\partial z_{i,t}}{\partial w_{11}} & \frac{\partial z_{i,t}}{\partial w_{12}} & \cdots & \frac{\partial z_{i,t}}{\partial w_{1M}} \\ \frac{\partial z_{i,t}}{\partial w_{21}} & \frac{\partial z_{i,t}}{\partial w_{22}} & \cdots & \frac{\partial z_{i,t}}{\partial w_{2M}} \\ \vdots & & & \\ \frac{\partial z_{i,t}}{\partial w_{M1}} & \frac{\partial z_{i,t}}{\partial w_{M2}} & \cdots & \frac{\partial z_{i,t}}{\partial w_{MM}} \end{pmatrix} := \begin{pmatrix} \frac{\partial z_{i,t}}{\partial \boldsymbol{w}_{1*}} \\ \frac{\partial z_{i,t}}{\partial \boldsymbol{w}_{2*}} \\ \vdots \\ \frac{\partial z_{i,t}}{\partial \boldsymbol{w}_{M*}} \end{pmatrix}, \qquad i = 1, 2, \cdots, M, \tag{15}$$

where $\boldsymbol{w}_{i*} \in \mathbb{R}^M$ is a row-vector.

Now, suppose that $\{\boldsymbol{z}_1, \boldsymbol{z}_2, \boldsymbol{z}_3, \ldots\}$ is an orbit of the system which converges to a stable fixed point, i.e. $\lim_{T\to\infty} \boldsymbol{z}_T = \boldsymbol{z}^{*k}$. Then

$$\lim_{T\to\infty} \boldsymbol{z}_T = \lim_{T\to\infty} \left(\boldsymbol{W}_{\Omega(T-1)} \boldsymbol{z}_{T-1} + \boldsymbol{h}\right) = \boldsymbol{z}^{*1} = \boldsymbol{W}_{\Omega(t^{*1})} \boldsymbol{z}^{*1} + \boldsymbol{h}, \tag{16}$$

and so

$$\lim_{T\to\infty} \left(\boldsymbol{W}_{\Omega(T-1)}\right) \boldsymbol{z}^{*1} = \boldsymbol{W}_{\Omega(t^{*1})} \boldsymbol{z}^{*1}. \tag{17}$$

Assume that $\lim_{T\to\infty} \left(\boldsymbol{W}_{\Omega(T-1)}\right) = \boldsymbol{L}$. Since eq. 17 holds for every $\boldsymbol{z}^{*1}$, then substituting $\boldsymbol{z}^{*1} = \boldsymbol{e}_1^T = (1, 0, \cdots, 0)^T$ in eq. 17, we can prove that the first column of $\boldsymbol{L}$ equals the first column of $\boldsymbol{W}_{\Omega(t^{*1})}$. Performing the same procedure for $\boldsymbol{z}^{*1} = \boldsymbol{e}_i^T$, $i = 2, 3, \cdots, M$, yields

$$\lim_{T\to\infty} \boldsymbol{W}_{\Omega(T-1)} = \boldsymbol{W}_{\Omega(t^{*1})}. \tag{18}$$

Also, for every $i \in \mathbb{N} \, (1 < i < \infty)$

$$\lim_{T\to\infty} \boldsymbol{W}_{\Omega(T-i)} = \boldsymbol{W}_{\Omega(t^{*1})}, \tag{19}$$

i.e.

$$\forall \epsilon > 0 \quad \exists N \in \mathbb{N} \quad s.t. \quad T - i \geq N \implies \left\|\boldsymbol{W}_{\Omega(T-i)} - \boldsymbol{W}_{\Omega(t^{*1})}\right\| \leq \epsilon. \tag{20}$$

Thus, $\left\|\boldsymbol{W}_{\Omega(T-i)}\right\| - \left\|\boldsymbol{W}_{\Omega(t^{*1})}\right\| \leq \left\|\boldsymbol{W}_{\Omega(T-i)} - \boldsymbol{W}_{\Omega(t^{*1})}\right\|$ gives

$$\forall \epsilon > 0 \quad \exists N \in \mathbb{N} \quad s.t. \quad T - i \geq N \implies \left\|\boldsymbol{W}_{\Omega(T-i)}\right\| \leq \left\|\boldsymbol{W}_{\Omega(t^{*1})}\right\| + \epsilon. \tag{21}$$

Since $T - 1 > T - 2 > \cdots > T - i \geq N$, so

$$\forall \epsilon > 0 \quad \left\|\boldsymbol{W}_{\Omega(T-i)}\right\| \leq \left\|\boldsymbol{W}_{\Omega(t^{*1})}\right\| + \epsilon, \quad i = 1, 2, \cdots, T - N. \tag{22}$$

Hence

$$\forall \epsilon > 0 \quad \left\|\prod_{i=1}^{T-N} \boldsymbol{W}_{\Omega(T-i)}\right\| \leq \prod_{i=1}^{T-N} \left\|\boldsymbol{W}_{\Omega(T-i)}\right\| \leq \left(\left\|\boldsymbol{W}_{\Omega(t^{*1})}\right\| + \epsilon\right)^{T-N}. \tag{23}$$

If $\left\|\boldsymbol{W}_{\Omega(t^{*1})}\right\| < 1$, then for any $\epsilon < 1$, considering $\bar{\epsilon} \leq \frac{\epsilon + \left\|\boldsymbol{W}_{\Omega(t^{*1})}\right\|}{2} < 1$, it is concluded that

$$\left\|\lim_{T\to\infty} \prod_{i=1}^{T-N} \boldsymbol{W}_{\Omega(T-i)}\right\| = \lim_{T\to\infty} \left\|\prod_{i=1}^{T-N} \boldsymbol{W}_{\Omega(T-i)}\right\| \leq \lim_{T\to\infty} \left(\left\|\boldsymbol{W}_{\Omega(t^{*1})}\right\| + \bar{\epsilon}\right)^{T-N} = 0. \tag{24}$$

Therefore

$$\lim_{T \to \infty} \prod_{i=1}^{T-1} \boldsymbol{W}_{\Omega(T-i)} = 0. \tag{25}$$

If the orbit $\{\boldsymbol{z}_1, \boldsymbol{z}_2, \boldsymbol{z}_3, \ldots\}$ tends to a stable $k$-cycle ($k > 1$) with the periodic points

$$\{F^k(\boldsymbol{z}^{*k}), F^{k-1}(\boldsymbol{z}^{*k}), F^{k-2}(\boldsymbol{z}^{*k}), \cdots, F(\boldsymbol{z}^{*k})\} = \{\boldsymbol{z}_{t^{*k}}, \boldsymbol{z}_{t^{*k}-1}, \cdots, \boldsymbol{z}_{t^{*k}-(k-1)}\},$$

then, denoting the stable $k$-cycle by

$$\Gamma_k = \{\boldsymbol{z}_{t^{*k}}, \boldsymbol{z}_{t^{*k}-1}, \cdots, \boldsymbol{z}_{t^{*k}-(k-1)}, \boldsymbol{z}_{t^{*k}}, \boldsymbol{z}_{t^{*k}-1}, \cdots, \boldsymbol{z}_{t^{*k}-(k-1)}, \cdots\}, \tag{26}$$

we have

$$\lim_{T \to \infty} d(\boldsymbol{z}_T, \Gamma_k) = 0. \tag{27}$$

Hence, there exists a neighborhood $U$ of $\Gamma_k$ and $k$ sub-sequences $\{\boldsymbol{z}_{T_{kn}}\}_{n=1}^{\infty}, \{\boldsymbol{z}_{T_{kn+1}}\}_{n=1}^{\infty}, \cdots,$ $\{\boldsymbol{z}_{T_{kn+(k-1)}}\}_{n=1}^{\infty}$ of the sequence $\{\boldsymbol{z}_T\}_{T=1}^{\infty}$ such that these sub-sequences belong to $U$ and

(i) $\boldsymbol{z}_{T_{kn+s}} = F^k(\boldsymbol{z}_{T_{k(n-1)+s}}), s = 0, 1, 2, \cdots, k-1,$

(ii) $\lim_{T \to \infty} \boldsymbol{z}_{T_{kn+s}} = \boldsymbol{z}_{t^{*k}-s}, s = 0, 1, 2, \cdots, k-1,$

(iii) for every $\boldsymbol{z}_T \in U$ there is some $s \in \{0, 1, 2, \cdots, k-1\}$ such that $\boldsymbol{z}_T \in \{\boldsymbol{z}_{T_{kn+s}}\}_{n=1}^{\infty}$.

In this case, for every $\boldsymbol{z}_T \in U$ with $\boldsymbol{z}_T \in \{\boldsymbol{z}_{T_{kn+s}}\}_{n=1}^{\infty}$ we have $\lim_{T \to \infty} \boldsymbol{z}_T = \boldsymbol{z}_{t^{*k}-s}$ for some $s = 0, 1, 2, \cdots, k-1$. Therefore, continuity of $F$ implies that $\lim_{T \to \infty} F(\boldsymbol{z}_T) = F(\boldsymbol{z}_{t^{*k}-s})$ and so

$$\lim_{T \to \infty} \left(\boldsymbol{W}_{\Omega(T)} \boldsymbol{z}_T + \boldsymbol{h}\right) = \boldsymbol{W}_{\Omega(t^{*k}-s)} \boldsymbol{z}_{t^{*k}-s} + \boldsymbol{h}. \tag{28}$$

Thus, similarly, we can prove that

$$\exists s \in \{0, 1, 2, \cdots, k-1\} \quad s.t. \quad \lim_{T \to \infty} \boldsymbol{W}_{\Omega(T)} = \boldsymbol{W}_{\Omega(t^{*k}-s)}. \tag{29}$$

Analogously, for every $i \in \mathbb{N} \, (1 < i < \infty)$

$$\exists s_i \in \{0, 1, 2, \cdots, k-1\} \quad s.t. \quad \lim_{T \to \infty} \boldsymbol{W}_{\Omega(T-i)} = \boldsymbol{W}_{\Omega(t^{*k}-s_i)}, \tag{30}$$

On the other hand, $\left\|\boldsymbol{W}_{\Omega(t^{*k}-s_i)}\right\| < 1$ for all $s_i \in \{0, 1, 2, \cdots, k-1\}$. So, without loss of generality, assuming

$$\max_{0 \leq s_i \leq k-1} \left\{\left\|\boldsymbol{W}_{\Omega(t^{*k}-s_i)}\right\|\right\} = \left\|\boldsymbol{W}_{\Omega(t^{*k})}\right\| < 1, \tag{31}$$

we can again obtain some relations similar to eq. 23-eq. 25 for $t^{*k}, k \geq 1$.

Since $\{\boldsymbol{z}_{T-1}\}_{T=1}^{\infty}$ is a convergent sequence, so it is bounded, i.e. there exists a real number $q > 0$ such that $\|\boldsymbol{z}_{T-1}\| \leq q$ for all $T \in \mathbb{N}$. Furthermore, $\left\|\boldsymbol{D}_{\Omega(T-1)}\right\| \leq 1$ for all $T$. Therefore, by eq. 12 and eq. 23 (for $t^{*k}, k \geq 1$)

$$\left\|\frac{\partial \boldsymbol{z}_T}{\partial w_{mk}}\right\|$$

$$= \left\|\mathbf{1}_{(m,k)} \boldsymbol{D}_{\Omega(T-1)} \boldsymbol{z}_{T-1} + \sum_{j=2}^{T-1} \left(\prod_{i=1}^{j-1} \boldsymbol{W}_{\Omega(T-i)}\right) \mathbf{1}_{(m,k)} \boldsymbol{D}_{\Omega(T-j)} \boldsymbol{z}_{T-j}\right.$$

$$\left. + \prod_{i=1}^{T-1} \boldsymbol{W}_{\Omega(T-i)} \boldsymbol{D}_{\Omega(1)} \boldsymbol{z}_1\right\| \tag{32}$$

$$\leq \|\boldsymbol{z}_{T-1}\| + \left[\sum_{j=2}^{T-1} \left\|\prod_{i=1}^{j-1} \boldsymbol{W}_{\Omega(T-i)}\right\| \|\boldsymbol{z}_{T-j}\|\right] + \left\|\prod_{i=1}^{T-1} \boldsymbol{W}_{\Omega(T-i)}\right\| \|\boldsymbol{z}_1\|$$

$$\leq q \left(1 + \sum_{j=2}^{T-1} \left(\left\|\boldsymbol{W}_{\Omega(t^{*k})}\right\| + \bar{\epsilon}\right)^{j-1}\right) + \left(\left\|\boldsymbol{W}_{\Omega(t^{*k})}\right\| + \bar{\epsilon}\right)^{T-1} \|\boldsymbol{z}_1\|. \tag{33}$$

Thus, by $\left\|\boldsymbol{W}_{\Omega(t*k)}\right\| + \bar{\epsilon} < 1$, we have

$$\lim_{T\to\infty} \left\|\frac{\partial \boldsymbol{z}_T}{\partial w_{mk}}\right\| \leq q\left(1 + \frac{\left\|\boldsymbol{W}_{\Omega(t*k)}\right\| + \bar{\epsilon}}{1 - \left\|\boldsymbol{W}_{\Omega(t*k)}\right\| - \bar{\epsilon}}\right) = \mathcal{M} < \infty, \tag{34}$$

i.e., by eq. 14 and eq. 15, the 2-norm of total gradient matrices and hence $\left\|\frac{\partial \boldsymbol{z}_t}{\partial \boldsymbol{W}}\right\|_2$ will not diverge (explode) under the assumptions of Theorem 1.

Analogously, we can prove that $\left\|\frac{\partial \boldsymbol{z}_T}{\partial \boldsymbol{A}}\right\|_2$ and $\left\|\frac{\partial \boldsymbol{z}_T}{\partial \boldsymbol{h}}\right\|_2$ will not diverge either. Since, similar as in the derivations above, it can be shown that relation eq. 34 is true for $\left\|\frac{\partial \boldsymbol{z}_T}{\partial a_{mm}}\right\|$ with $q = \bar{q}$, where $\bar{q}$ is the upper bound of $\|\boldsymbol{z}_T\|$, as $\{\boldsymbol{z}_T\}_{T=1}^{\infty}$ is convergent. Furthermore, relation eq. 34 also holds for $\left\|\frac{\partial \boldsymbol{z}_T}{\partial h_m}\right\|$ with $q = 1$.

**Remark 2.1.** *By eq. 24 the Jacobian parts $\left\|\frac{\partial \boldsymbol{z}_T}{\partial \boldsymbol{z}_t}\right\|_2$ connecting any two states $\boldsymbol{z}_T$ and $\boldsymbol{z}_t$, $T > t$, will not diverge either.*

**Corollary 2.1.** *The results of Theorem 1 are also true if $\boldsymbol{W}_{\Omega(t*k)}$ is a normal matrix with no eigenvalue equal to one.*

*Proof.* If $\boldsymbol{W}_{\Omega(t*k)}$ is normal, then $\left\|\boldsymbol{W}_{\Omega(t*k)}\right\| = \rho(\boldsymbol{W}_{\Omega(t*k)}) < 1$ which satisfies the conditions of Theorem 1. □

### 6.1.4 *Proof of Theorem 2*

Let $\boldsymbol{A}$, $\boldsymbol{W}$ and $\boldsymbol{D}_{\Omega(k)}$, $t < k \leq T$, be partitioned as follows

$$\boldsymbol{A} = \left(\begin{array}{c|c} \boldsymbol{I}_{reg} & \boldsymbol{O}^{\mathsf{T}} \\ \hline \boldsymbol{O} & \boldsymbol{A}_{nreg} \end{array}\right), \qquad \boldsymbol{W} = \left(\begin{array}{c|c} \boldsymbol{O}_{reg} & \boldsymbol{O}^{\mathsf{T}} \\ \hline \boldsymbol{S} & \boldsymbol{W}_{nreg} \end{array}\right), \qquad \boldsymbol{D}_{\Omega(k)} = \left(\begin{array}{c|c} \boldsymbol{D}^k_{reg} & \boldsymbol{O}^{\mathsf{T}} \\ \hline \boldsymbol{O} & \boldsymbol{D}^k_{nreg} \end{array}\right), \tag{35}$$

where $\boldsymbol{I}_{M_{reg}\times M_{reg}} := \boldsymbol{I}_{reg} \in \mathbb{R}^{M_{reg}\times M_{reg}}, \boldsymbol{O}_{M_{reg}\times M_{reg}} := \boldsymbol{O}_{reg} \in \mathbb{R}^{M_{reg}\times M_{reg}}, \boldsymbol{O}, \boldsymbol{S} \in \mathbb{R}^{(M-M_{reg})\times M_{reg}}, \boldsymbol{A}_{\{M_{reg}+1:M,M_{reg}+1:M\}} := \boldsymbol{A}_{nreg} \in \mathbb{R}^{(M-M_{reg})\times(M-M_{reg})}$ is a diagonal sub-matrix, $\boldsymbol{W}_{\{M_{reg}+1:M,M_{reg}+1:M\}} := \boldsymbol{W}_{nreg} \in \mathbb{R}^{(M-M_{reg})\times(M-M_{reg})}$ is an off-diagonal sub-matrix (cf. Fig. S1). Moreover, $\boldsymbol{D}^k_{M_{reg}\times M_{reg}} := \boldsymbol{D}^k_{reg} \in \mathbb{R}^{M_{reg}\times M_{reg}}$ and $\boldsymbol{D}^k_{\{M_{reg}+1:M,M_{reg}+1:M\}} := \boldsymbol{D}^k_{nreg} \in \mathbb{R}^{(M-M_{reg})\times(M-M_{reg})}$ are diagonal sub-matrices. Then, we have

$$\prod_{t<k\leq T} W_{\Omega(k)}$$

$$= \prod_{t<k\leq T} \left(\begin{array}{c|c} \boldsymbol{I}_{reg} & \boldsymbol{O}^{\mathsf{T}} \\ \hline \boldsymbol{S}\,\boldsymbol{D}^k_{reg} & \boldsymbol{A}_{nreg} + \boldsymbol{W}_{nreg}\,\boldsymbol{D}^k_{nreg} \end{array}\right) := \prod_{t<k\leq T} \left(\begin{array}{c|c} \boldsymbol{I}_{reg} & \boldsymbol{O}^{\mathsf{T}} \\ \hline \boldsymbol{S}\,\boldsymbol{D}^k_{reg} & \boldsymbol{W}^k_{nreg} \end{array}\right)$$

$$= \left(\begin{array}{c|c} \boldsymbol{I}_{reg} & \boldsymbol{O}^{\mathsf{T}} \\ \hline \boldsymbol{S}\,\boldsymbol{D}^{t+1}_{reg} + \sum_{j=2}^{T}\left(\prod_{t<k\leq t+j-1}\boldsymbol{W}^k_{nreg}\right)\boldsymbol{S}\,\boldsymbol{D}^{t+j}_{reg} & \prod_{t<k\leq T}\boldsymbol{W}^k_{nreg}. \end{array}\right) \tag{36}$$

Therefore, considering the 2-norm, we obtain

$$\left\|\frac{\partial \boldsymbol{z}_T}{\partial \boldsymbol{z}_t}\right\| = \left\|\prod_{t<k\leq T} \boldsymbol{W}_{\Omega(k)}\right\|$$

$$= \left\|\left(\begin{array}{c|c} \boldsymbol{I}_{reg} & \boldsymbol{O}^{\mathsf{T}} \\ \hline \boldsymbol{S}\,\boldsymbol{D}^{t+1}_{reg} + \sum_{j=2}^{T}\left(\prod_{t<k\leq t+j-1}\boldsymbol{W}^k_{nreg}\right)\boldsymbol{S}\,\boldsymbol{D}^{t+j}_{reg} & \prod_{t<k\leq T}\boldsymbol{W}^k_{nreg} \end{array}\right)\right\| < \infty. \tag{37}$$

Moreover

$$1 \leq \max\{1, \rho(W_{T-t})\} = \rho\big( \prod_{t<k\leq T} W_{\Omega(k)} \big) \leq \left\| \prod_{t<k\leq T} \boldsymbol{W}_{\Omega(k)} \right\| = \left\| \frac{\partial \boldsymbol{z}_T}{\partial \boldsymbol{z}_t} \right\| \quad (38)$$

where $W_{T-t} := \prod_{t<k\leq T} \boldsymbol{W}_{nreg}^k$. Therefore, eq. 37 and eq. 38 yield

$$1 \leq \rho_{low} \leq \left\| \frac{\partial \boldsymbol{z}_T}{\partial \boldsymbol{z}_t} \right\| \leq \rho_{up} < \infty.$$

Furthermore, we assumed that the non-regularized subsystem $(z_{M_{reg}+1} \ldots z_M)$, if considered in isolation, satisfies Theorem 1. Hence, similar to the proof of Theorem 1, it is concluded that

$$\lim_{T\to\infty} \prod_{k=t}^{T} \boldsymbol{W}_{nreg}^k = \boldsymbol{O}_{nreg}. \quad (39)$$

On the other hand, by definition of $\boldsymbol{D}_{\Omega(k)}$, for every $t < k \leq T$, we have $\left\| \boldsymbol{D}_{reg}^k \right\| \leq 1$ and so

$$\left\| \boldsymbol{S}\,\boldsymbol{D}_{reg}^k \right\| \leq \|\boldsymbol{S}\| \left\| \boldsymbol{D}_{reg}^k \right\| \leq \|\boldsymbol{S}\|, \quad (40)$$

which, in accordance with the the assumptions of Theorem 1, by convergence of $\sum_{j=2}^{T} \prod_{k=t+1}^{t+j-1} \left\| \boldsymbol{W}_{nreg}^k \right\|$ implies

$$\lim_{T\to\infty} \left\| \boldsymbol{S}\,\boldsymbol{D}_{reg}^{t+1} + \sum_{j=2}^{T} \big( \prod_{k=t+1}^{t+j-1} \boldsymbol{W}_{nreg}^k \big) \boldsymbol{S}\,\boldsymbol{D}_{reg}^{t+j} \right\| \leq \|\boldsymbol{S}\| \left( 1 + \lim_{T\to\infty} \sum_{j=2}^{T} \prod_{k=t+1}^{t+j-1} \left\| \boldsymbol{W}_{nreg}^k \right\| \right)$$

$$\leq \|\boldsymbol{S}\|\,\mathcal{M}_{nreg}. \quad (41)$$

Thus, denoting $\boldsymbol{Q} := \boldsymbol{S}\,\boldsymbol{D}_{reg}^{t+1} + \sum_{j=2}^{T} \big( \prod_{t<k\leq t+j-1} \boldsymbol{W}_{nreg}^k \boldsymbol{S}\,\boldsymbol{D}_{reg}^{t+j} \big)$, from eq. 41 we deduce that

$$\lambda_{max}\big( \lim_{T\to\infty} (\boldsymbol{Q}^{\mathsf{T}}\boldsymbol{Q}) \big) = \lim_{T\to\infty} \rho(\boldsymbol{Q}^{\mathsf{T}}\boldsymbol{Q}) \leq \lim_{T\to\infty} \left\| \boldsymbol{Q}^{\mathsf{T}}\boldsymbol{Q} \right\| = \lim_{T\to\infty} \|\boldsymbol{Q}\|^2 \leq \big( \|\boldsymbol{S}\|\,\mathcal{M}_{nreg} \big)^2. \quad (42)$$

Now, if $T - t$ tends to $\infty$, then eq. 37, eq. 39 and eq. 42 result in

$$1 = \rho_{low} \leq \left\| \frac{\partial \boldsymbol{z}_T}{\partial \boldsymbol{z}_t} \right\| = \sigma_{max}\left( \left( \begin{array}{c|c} \boldsymbol{I}_{reg} & \boldsymbol{O}^{\mathsf{T}} \\ \hline \boldsymbol{Q} & \boldsymbol{O}_{nreg} \end{array} \right) \right) = \sqrt{\lambda_{max}(\boldsymbol{I}_{reg} + \lim_{T\to\infty} (\boldsymbol{Q}^{\mathsf{T}}\boldsymbol{Q}))}$$

$$= \rho_{up} < \infty. \quad (43)$$

**Remark 2.2.** *If* $\|\boldsymbol{S}\| = 0$*, then* $\left\| \frac{\partial \boldsymbol{z}_T}{\partial \boldsymbol{z}_t} \right\| \to 1$ *as* $T - t \to \infty$.

### 6.1.5 *Details on EM algorithm and DS reconstruction*

For DS reconstruction we request that the latent RNN approximates the true generating system of equations, which is a taller order than learning the mapping $\boldsymbol{S} \to \boldsymbol{X}$ or predicting future values in a time series (cf. sect. 3.5).[2] This point has important implications for the design of models, inference algorithms and performance metrics if the primary goal is DS reconstruction rather than 'mere' time series forecasting.[3] In this context we consider the fully probabilistic, generative RNN eq. 1. Together with eq. 2 (where we take $g(\boldsymbol{z}_t) = \phi(\boldsymbol{z}_t)$) this gives the typical form of a nonlinear

---

[2]By reconstructing the governing equations we mean their approximation in the sense of the universal approximation theorems for DS (Funahashi & Nakamura, 1993; Kimura & Nakano, 1998), i.e. such that the behavior of the reconstructed system becomes *dynamically* equivalent to that of the true underlying system.

[3]In this context we also remark that models which include longer histories of hidden activations (Yu et al., 2019), as in many statistical time series models (Fan & Yao, 2003), are not formally valid DS models anymore since they violate the uniqueness of flow in state space (Strogatz, 2015).

state space model (Durbin & Koopman, 2012) with observation and process noise. We solve for the parameters $\boldsymbol{\theta} = \{\boldsymbol{A}, \boldsymbol{W}, \boldsymbol{C}, \boldsymbol{h}, \boldsymbol{\mu}_0, \boldsymbol{\Sigma}, \boldsymbol{B}, \boldsymbol{\Gamma}\}$ by maximum likelihood, for which an efficient Expectation-Maximization (EM) algorithm has recently been suggested (Durstewitz, 2017; Koppe et al., 2019), which we will summarize here. Since the involved integrals are not tractable, we start off from the evidence-lower bound (ELBO) to the log-likelihood which can be rewritten in various useful ways:

$$
\begin{aligned}
\log p(\boldsymbol{X}|\boldsymbol{\theta}) &\geq \mathbb{E}_{\boldsymbol{Z}\sim q}[\log p_{\boldsymbol{\theta}}(\boldsymbol{X}, \boldsymbol{Z})] + H\left(q(\boldsymbol{Z}|\boldsymbol{X})\right) \\
&= \log p(\boldsymbol{X}|\boldsymbol{\theta}) - D_{\mathrm{KL}}\left(q(\boldsymbol{Z}|\boldsymbol{X})\|p_{\boldsymbol{\theta}}(\boldsymbol{Z}|\boldsymbol{X})\right) \\
&=: \mathcal{L}\left(\boldsymbol{\theta}, q\right)
\end{aligned}
\tag{44}
$$

In the E-step, given a current estimate $\boldsymbol{\theta}^*$ for the parameters, we seek to determine the posterior $p_{\boldsymbol{\theta}}\left(\boldsymbol{Z}|\boldsymbol{X}\right)$ which we approximate by a global Gaussian $q(\boldsymbol{Z}|\boldsymbol{X})$ instantiated by the maximizer (mode) $\boldsymbol{Z}^*$ of $p_{\boldsymbol{\theta}}(\boldsymbol{Z}|\boldsymbol{X})$ as an estimator of the mean, and the negative inverse Hessian around this maximizer as an estimator of the state covariance, i.e.

$$
\begin{aligned}
\mathbb{E}[\boldsymbol{Z}|\boldsymbol{X}] \approx \boldsymbol{Z}^* &= \arg\max_{\boldsymbol{Z}} \log p_{\boldsymbol{\theta}}(\boldsymbol{Z}|\boldsymbol{X}) \\
&= \arg\max_{\boldsymbol{Z}}[\log p_{\boldsymbol{\theta}}(\boldsymbol{X}|\boldsymbol{Z}) + \log p_{\boldsymbol{\theta}}(\boldsymbol{Z}) - \log p_{\boldsymbol{\theta}}(\boldsymbol{X})] \\
&= \arg\max_{\boldsymbol{Z}} \left[\log p_{\boldsymbol{\theta}}(\boldsymbol{X}|\boldsymbol{Z}) + \log p_{\boldsymbol{\theta}}(\boldsymbol{Z})\right],
\end{aligned}
\tag{45}
$$

since $\boldsymbol{Z}$ integrates out in $p_{\boldsymbol{\theta}}(\boldsymbol{X})$ (equivalently, this result can be derived from a Laplace approximation to the log-likelihood, $\log p(\boldsymbol{X}|\boldsymbol{\theta}) \approx \log p_{\boldsymbol{\theta}}(\boldsymbol{X}|\boldsymbol{Z}^*) + \log p_{\boldsymbol{\theta}}(\boldsymbol{Z}^*) - \frac{1}{2}\log|-\boldsymbol{L}^*| + \mathrm{const}$, where $\boldsymbol{L}^*$ is the Hessian evaluated at the maximizer). We solve this optimization problem by a fixed-point iteration scheme that efficiently exploits the model's piecewise linear structure, as detailed below.

Using this approximate posterior for $p_{\boldsymbol{\theta}}(\boldsymbol{Z}|\boldsymbol{X})$, based on the model's piecewise-linear structure most of the expectation values $\mathbb{E}_{\boldsymbol{z}\sim q}\left[\phi(\boldsymbol{z})\right]$, $\mathbb{E}_{\boldsymbol{z}\sim q}\left[\phi(\boldsymbol{z})\boldsymbol{z}^{\mathsf{T}}\right]$, and $\mathbb{E}_{\boldsymbol{z}\sim q}\left[\phi(\boldsymbol{z})\phi(\boldsymbol{z})^{\mathsf{T}}\right]$, could be solved for (semi-)analytically (where $\boldsymbol{z}$ is the concatenated vector form of $\boldsymbol{Z}$, see below). In the M-step, we seek $\boldsymbol{\theta}^* := \arg\max_{\boldsymbol{\theta}} \mathcal{L}(\boldsymbol{\theta}, q^*)$, assuming proposal density $q^*$ to be given from the E-step, which for a Gaussian observation model amounts to a simple linear regression problem (see Suppl. eq. 49). To force the PLRNN to really capture the underlying DS in its governing equations, we use a previously suggested (Koppe et al., 2019) stepwise annealing protocol that gradually shifts the burden of fitting the observations $\boldsymbol{X}$ from the observation model eq. 2 to the latent RNN model eq. 1 during training, the idea of which is to establish a mapping from latent states $\boldsymbol{Z}$ to observations $\boldsymbol{X}$ first, fixing this, and then enforcing the temporal consistency constraints implied by eq. 1 while accounting for the actual observations.

Now we briefly outline the fixed-point-iteration algorithm for solving the maximization problem in eq. 45 (for more details see Durstewitz (2017); Koppe et al. (2019)). Given a Gaussian latent PLRNN and a Gaussian observation model, the joint density $p(\boldsymbol{X}, \boldsymbol{Z})$ will be piecewise Gaussian, hence eq. 45 piecewise quadratic in $\boldsymbol{Z}$. Let us concatenate all state variables across $m$ and $t$ into one long column vector $\boldsymbol{z} = (z_{1,1}, \ldots, z_{M,1}, \ldots, z_{1,T}, \ldots, z_{M,T})^{\mathsf{T}}$, arrange matrices $\boldsymbol{A}$, $\boldsymbol{W}$ into large $MT \times MT$ block tri-diagonal matrices, define $\boldsymbol{d}_{\Omega} := \left(\mathbf{1}_{z_{1,1}>0}, \mathbf{1}_{z_{2,1}>0}, \ldots, \mathbf{1}_{z_{M,T}>0}\right)^{\mathsf{T}}$ as an indicator vector with a 1 for all states $z_{m,t} > 0$ and zeros otherwise, and $\boldsymbol{D}_{\Omega} := \mathrm{diag}(\boldsymbol{d}_{\Omega})$ as the diagonal matrix formed from this vector. Collecting all terms quadratic, linear, or constant in $\boldsymbol{z}$, we can then write down the optimization criterion in the form

$$
\begin{aligned}
Q_{\Omega}^*(\boldsymbol{z}) = -\frac{1}{2}[&\boldsymbol{z}^{\mathsf{T}}\left(\boldsymbol{U}_0 + \boldsymbol{D}_{\Omega}\boldsymbol{U}_1 + \boldsymbol{U}_1^{\mathsf{T}}\boldsymbol{D}_{\Omega} + \boldsymbol{D}_{\Omega}\boldsymbol{U}_2\boldsymbol{D}_{\Omega}\right)\boldsymbol{z} \\
&- \boldsymbol{z}^{\mathsf{T}}\left(\boldsymbol{v}_0 + \boldsymbol{D}_{\Omega}\boldsymbol{v}_1\right) - \left(\boldsymbol{v}_0 + \boldsymbol{D}_{\Omega}\boldsymbol{v}_1\right)^{\mathsf{T}}\boldsymbol{z}] + \mathrm{const}.
\end{aligned}
\tag{46}
$$

In essence, the algorithm now iterates between the two steps:

1. Given fixed $\boldsymbol{D}_{\Omega}$, solve

$$
\boldsymbol{z}^* = \left(\boldsymbol{U}_0 + \boldsymbol{D}_{\Omega}\boldsymbol{U}_1 + \boldsymbol{U}_1^{\mathsf{T}}\boldsymbol{D}_{\Omega} + \boldsymbol{D}_{\Omega}\boldsymbol{U}_2\boldsymbol{D}_{\Omega}\right)^{-1} \cdot \left(\boldsymbol{v}_0 + \boldsymbol{D}_{\Omega}\boldsymbol{v}_1\right)
\tag{47}
$$

2. Given fixed $\boldsymbol{z}^*$, recompute $\boldsymbol{D}_{\Omega}$

until either convergence or one of several stopping criteria (partly likelihood-based, partly to avoid loops) is reached. The solution may afterwards be refined by one quadratic programming step. Numerical experiments showed this algorithm to be very fast and efficient (Durstewitz, 2017; Koppe et al., 2019). At $\boldsymbol{z}^*$, an estimate of the state covariance is then obtained as the inverse negative Hessian,

$$\boldsymbol{V} = \left(\boldsymbol{U}_0 + \boldsymbol{D}_\Omega \boldsymbol{U}_1 + \boldsymbol{U}_1^\mathsf{T} \boldsymbol{D}_\Omega + \boldsymbol{D}_\Omega \boldsymbol{U}_2 \boldsymbol{D}_\Omega\right)^{-1}. \tag{48}$$

In the M-step, using the proposal density $q^*$ from the E-step, the solution to the maximization problem $\boldsymbol{\theta}^* \coloneqq \arg\max_{\boldsymbol{\theta}} \mathcal{L}(\boldsymbol{\theta}, q^*)$, can generally be expressed in the form

$$\boldsymbol{\theta}^* = \left(\sum_t \mathbb{E}\left[\boldsymbol{\alpha}_t \boldsymbol{\beta}_t^\mathsf{T}\right]\right) \left(\sum_t \mathbb{E}\left[\boldsymbol{\beta}_t \boldsymbol{\beta}_t^\mathsf{T}\right]\right)^{-1}, \tag{49}$$

where, for the latent model, eq. 1, $\boldsymbol{\alpha}_t = \boldsymbol{z}_t$ and $\boldsymbol{\beta}_t \coloneqq \left[\boldsymbol{z}_{t-1}^\mathsf{T}, \phi(\boldsymbol{z}_{t-1})^\mathsf{T}, \boldsymbol{s}_t^\mathsf{T}, 1\right]^\mathsf{T} \in \mathbb{R}^{2M+K+1}$, and for the observation model, eq. 2, $\boldsymbol{\alpha}_t = \boldsymbol{x}_t$ and $\boldsymbol{\beta}_t = g(\boldsymbol{z}_t)$.

### 6.1.6 *More details on DS performance measure*

As argued before (Koppe et al., 2019; Wood, 2010), in DS reconstruction we require that the RNN captures the underlying *attractor geometries* and state space properties. This does not necessarily entail that the reconstructed system could predict future time series observations more than a few time steps ahead, and vice versa. For instance, if the underlying attractor is chaotic, even if we had the *exact true system* available, with a tiny bit of noise trajectories starting from the same initial condition will quickly diverge and ahead-prediction errors become essentially meaningless as a DS performance metric (Fig. S2**B**).

To quantify how well an inferred PLRNN captured the underlying dynamics we therefore followed Koppe et al. (2019) and used the Kullback-Leibler divergence between the true and reproduced probability distributions across states in state space, thus assessing the agreement in attractor geometries (cf. Takens (1981); Sauer et al. (1991)) rather than in precise matching of time series,

$$D_{\mathrm{KL}}\left(p_{\mathrm{true}}(\boldsymbol{x}) \| p_{\mathrm{gen}}(\boldsymbol{x}|\boldsymbol{z})\right) \approx \sum_{k=1}^K \hat{p}_{\mathrm{true}}^{(k)}(\boldsymbol{x}) \log\left(\frac{\hat{p}_{\mathrm{true}}^{(k)}(\boldsymbol{x})}{\hat{p}_{\mathrm{gen}}^{(k)}(\boldsymbol{x}|\boldsymbol{z})}\right), \tag{50}$$

where $p_{\mathrm{true}}(\boldsymbol{x})$ is the true distribution of observations across state space (not time!), $p_{\mathrm{gen}}(\boldsymbol{x}|\boldsymbol{z})$ is the distribution of observations generated by running the inferred PLRNN, and the sum indicates a spatial discretization (binning) of the observed state space. We emphasize that $\hat{p}_{\mathrm{gen}}^{(k)}(\boldsymbol{x}|\boldsymbol{z})$ is obtained from freely *simulated* trajectories, i.e. drawn from the prior $\hat{p}(\boldsymbol{z})$ specified by eq. 1, not from the inferred posteriors $\hat{p}(\boldsymbol{z}|\boldsymbol{x}_{\mathrm{train}})$. In addition, to assess reproduction of time scales by the inferred PLRNN, the average MSE between the power spectra of the true and generated time series was computed, as displayed in Fig. 3**B**–**C**.

The measure $D_{\mathrm{KL}}$ introduced above only works for situations where the ground truth $p_{\mathrm{true}}(\boldsymbol{X})$ is known. Following Koppe et al. (2019), we next briefly indicate how a proxy for $D_{\mathrm{KL}}$ may be obtained in empirical situations where no ground truth is available. Reasoning that for a well reconstructed DS the inferred posterior $p_{\mathrm{inf}}(\boldsymbol{z}|\boldsymbol{x})$ given the observations should be a good representative of the prior generative dynamics $p_{\mathrm{gen}}(\boldsymbol{z})$, one may use the Kullback-Leibler divergence between the distribution over latent states, obtained by sampling from the prior density $p_{\mathrm{gen}}(\boldsymbol{z})$, and the (data-constrained) posterior distribution $p_{\mathrm{inf}}(\boldsymbol{z}|\boldsymbol{x})$ (where $\boldsymbol{z} \in \mathbb{R}^{M \times 1}$ and $\boldsymbol{x} \in \mathbb{R}^{N \times 1}$), taken across the system's state space:

$$D_{\mathrm{KL}}\left(p_{\mathrm{inf}}(\boldsymbol{z}|\boldsymbol{x}) \| p_{\mathrm{gen}}(\boldsymbol{z})\right) = \int_{\boldsymbol{z} \in \mathbb{R}^{M \times 1}} p_{\mathrm{inf}}(\boldsymbol{z}|\boldsymbol{x}) \log \frac{p_{\mathrm{inf}}(\boldsymbol{z}|\boldsymbol{x})}{p_{\mathrm{gen}}(\boldsymbol{z})} d\boldsymbol{z} \tag{51}$$

As evaluating this integral is difficult, one could further approximate $p_{\mathrm{inf}}(\boldsymbol{z}|\boldsymbol{x})$ and $p_{\mathrm{gen}}(\boldsymbol{z})$ by Gaussian mixtures across trajectories, i.e. $p_{\mathrm{inf}}(\boldsymbol{z}|\boldsymbol{x}) \approx \frac{1}{T}\sum_{t=1}^T p(\boldsymbol{z}_t|\boldsymbol{x}_{1:T})$ and $p_{\mathrm{gen}}(\boldsymbol{z}) \approx \frac{1}{L}\sum_{l=1}^L p(\boldsymbol{z}_l|\boldsymbol{z}_{l-1})$, where the mean and covariance of $p(\boldsymbol{z}_t|\boldsymbol{x}_{1:T})$ and $p(\boldsymbol{z}_l|\boldsymbol{z}_{l-1})$ are obtained by marginalizing over the multivariate distributions $p(\boldsymbol{Z}|\boldsymbol{X})$ and $p_{\mathrm{gen}}(\boldsymbol{Z})$, respectively, yielding $\mathbb{E}[\boldsymbol{z}_t|\boldsymbol{x}_{1:T}]$, $\mathbb{E}[\boldsymbol{z}_l|\boldsymbol{z}_{l-1}]$, and covariance matrices $\mathrm{Var}(\boldsymbol{z}_t|\boldsymbol{x}_{1:T})$ and $\mathrm{Var}(\boldsymbol{z}_l|\boldsymbol{z}_{l-1})$. Supplementary

eq. 51 may then be numerically approximated through Monte Carlo sampling (Hershey & Olsen, 2007) by

$$D_{\mathrm{KL}}\left(p_{\mathrm{inf}}(\boldsymbol{z}|\boldsymbol{x})\|p_{\mathrm{gen}}(\boldsymbol{z})\right) \approx \frac{1}{n}\sum_{i=1}^{n}\log\frac{p_{\mathrm{inf}}(\boldsymbol{z}^{(i)}|\boldsymbol{x})}{p_{\mathrm{gen}}(\boldsymbol{z}^{(i)})},$$

$$\boldsymbol{z}^{(i)} \sim p_{\mathrm{inf}}(\boldsymbol{z}|\boldsymbol{x}) \tag{52}$$

Alternatively, there is also a variational approximation of eq. 51 available (Hershey & Olsen, 2007):

$$D_{\mathrm{KL}}^{\mathrm{variational}}\left(p_{\mathrm{inf}}(\boldsymbol{z}|\boldsymbol{x})\|p_{\mathrm{gen}}(\boldsymbol{z})\right) \approx \frac{1}{T}\sum_{t=1}^{T}\log\frac{\sum_{j=1}^{T}e^{-D_{\mathrm{KL}}(p(\boldsymbol{z}_t|\boldsymbol{x}_{1:T})\|p(\boldsymbol{z}_j|\boldsymbol{x}_{1:T}))}}{\sum_{k=1}^{T}e^{-D_{\mathrm{KL}}(p(\boldsymbol{z}_t|\boldsymbol{x}_{1:T})\|p(\boldsymbol{z}_k|\boldsymbol{z}_{k-1}))}}, \tag{53}$$

where the KL divergences in the exponentials are among Gaussians for which we have an analytical expression.

### 6.1.7 *More details on benchmark tasks and model comparisons*

We compared the performance of our rPLRNN to the other models summarized in Suppl. Table 1 on the following three benchmarks requiring long short-term maintenance of information (Talathi & Vartak (2016); Hochreiter & Schmidhuber (1997)): **1)** The *addition problem* of time length $T$ consists of $100\,000$ training and $10\,000$ test samples of $2 \times T$ input series $\boldsymbol{S} = \{\boldsymbol{s}_1, \ldots, \boldsymbol{s}_T\}$, where entries $\boldsymbol{s}_{1,:} \in [0,1]$ are drawn from a uniform random distribution and $\boldsymbol{s}_{2,:} \in \{0,1\}$ contains zeros except for two indicator bits placed randomly at times $t_1 < 10$ and $t_2 < T/2$. Constraints on $t_1$ and $t_2$ are chosen such that every trial requires a long memory of at least $T/2$ time steps. At the last time step $T$, the target output of the network is the sum of the two inputs in $\boldsymbol{s}_{1,:}$ indicated by the 1-entries in $\boldsymbol{s}_{2,:}$, $x_T^{\mathrm{target}} = s_{1,t_1} + s_{1,t_2}$. **2)** The *multiplication problem* is the same as the addition problem, only that the product instead of the sum has to be produced by the RNN as an output at time $T$, $x_T^{\mathrm{target}} = s_{1,t_1} \cdot s_{1,t_2}$. **3)** The MNIST dataset (LeCun et al., 2010) consists of $60\,000$ training and $10\,000$ $28 \times 28$ test images of hand written digits. To make this a time series problem, in *sequential MNIST* the images are presented sequentially, pixel-by-pixel, scanning lines from upper left to bottom-right, resulting in time series of fixed length $T = 784$.

For training on the addition and multiplication problems, the mean squared-error loss across $R$ samples, $\mathcal{L} = \frac{1}{R}\sum_{n=1}^{R}\left(\hat{x}_T^{(n)} - x_T^{(n)}\right)^2$, between estimated and actual outputs was used, while the cross-entropy loss $\mathcal{L} = \sum_{n=1}^{R}\left(-\sum_{i=1}^{10}x_{i,T}^{(n)}\log(\hat{p}_{i,T}^{(n)})\right)$ was employed for sequential MNIST, where

$$\hat{p}_{i,t} := \hat{p}_t\left(x_{i,t} = 1|\boldsymbol{z}_t\right) = \left(e^{\boldsymbol{B}_{i,:}\boldsymbol{z}_t}\right)\left(\sum_{j=1}^{N}e^{\boldsymbol{B}_{j,:}\boldsymbol{z}_t}\right)^{-1}, \tag{54}$$

with $x_{i,t} \in \{0,1\}$, $\sum_i x_{i,t} = 1$. We remark that as long as the observation model takes the form of a *generalized linear model* (Fahrmeir & Tutz, 2001), as assumed here, meaning may be assigned to the latent states $z_m$ by virtue of their association with specific sets of observations $x_n$ through the factor loading matrix $\boldsymbol{B}$. This adds another layer of model interpretability (besides its accessibility in DS terms).

The large error bars in Fig. 2 at the transition from good to bad performance result from the fact that the networks mostly learn these tasks in an all-or-none fashion. While the rPLRNN in general outperformed the pure initialization-based models (iRNN, npRNN, iPLRNN), confirming that a manifold attractor subspace present at initialization may be lost throughout training, we conjecture that this difference in performance will become even more pronounced as noise levels or task complexity increase.

Table 1: Overview over the different models used for comparison

| NAME | DESCRIPTION |
| --- | --- |
| RNN | Vanilla ReLU based RNN |
| iRNN | RNN with initialization $\boldsymbol{W}_0 = \boldsymbol{I}$ and $\boldsymbol{h}_0 = \boldsymbol{0}$ (Le et al., 2015) |
| npRNN | RNN with weights initialized to a normalized positive definite matrix with largest eigenvalue of 1 and biases initialized to zero (Talathi & Vartak, 2016) |
| PLRNN | PLRNN as given in eq. 1 (Koppe et al., 2019) |
| iPLRNN | PLRNN with initialization $\boldsymbol{A}_0 = \boldsymbol{I}$, $\boldsymbol{W}_0 = \boldsymbol{0}$ and $\boldsymbol{h}_0 = \boldsymbol{0}$ |
| rPLRNN | PLRNN initialized as illustrated in Fig. S1, with additional regularization term (eq. 3) |
| LSTM | Long Short-Term Memory (Hochreiter & Schmidhuber, 1997) |
| oRNN | ReLU RNN with $\boldsymbol{W}$ regularized toward orthogonality ($\boldsymbol{W}\boldsymbol{W}^T \rightarrow \mathbf{I}$) (Vorontsov et al., 2017) |
| L2RNN | Vanilla RNN with standard L2 regularization on all weights |
| L2pPLRNN | PLRNN with (partial) standard L2 regularization for proportion $M_{\text{reg}}/M = 0.5$ of units (i.e., pushing all terms in $\boldsymbol{A}$ and $\boldsymbol{W}$ for these units to 0) |
| L2fPLRNN | PLRNN with (full) standard L2 regularization on all weights |

### 6.1.8 *More details on single neuron model*

The neuron model used in section 4.2 is described by

$$
\begin{aligned}
-C_m \dot{V} = {} & g_L(V - E_L) + g_{Na} m_\infty(V)(V - E_{Na}) \\
& + g_K n(V - E_K) + g_M h(V - E_K) \\
& + g_{NMDA} \sigma(V)(V - E_{NMDA})
\end{aligned}
\tag{55}
$$

$$
\dot{h} = \frac{h_\infty(V) - h}{\tau_h}
\tag{56}
$$

$$
\dot{n} = \frac{n_\infty(V) - n}{\tau_n}
\tag{57}
$$

$$
\sigma(V) = \left[ 1 + .33 e^{-.0625V} \right]^{-1}
\tag{58}
$$

where $C_m$ refers to the neuron's membrane capacitance, the $g_\bullet$ to different membrane conductances, $E_\bullet$ to the respective reversal potentials, and $m$, $h$, and $n$ are gating variables with limiting values given by

$$
\{m_\infty, n_\infty, h_\infty\} = \left[ 1 + e^{(\{V_{hNa}, V_{hK}, V_{hM}\} - V)/\{k_{Na}, k_K, k_M\}} \right]^{-1}
\tag{59}
$$

Different parameter settings in this model lead to different dynamical phenomena, including regular spiking, slow bursting or chaos (see Durstewitz (2009) for details). Parameter settings used here were: $C_m = 6\,\mu\text{F}$, $g_L = 8\,\text{mS}$, $E_L = -80\,\text{mV}$, $g_{Na} = 20\,\text{mS}$, $E_{Na} = 60\,\text{mV}$, $V_{hNa} = -20\,\text{mV}$, $k_{Na} = 15$, $g_K = 10\,\text{mS}$, $E_K = -90\,\text{mV}$, $V_{hK} = -25\,\text{mV}$, $k_K = 5$, $\tau_n = 1\,\text{ms}$, $g_M = 25\,\text{mS}$, $V_{hM} = -15\,\text{mV}$, $k_M = 5$, $\tau_h = 200\,\text{ms}$, $g_{NMDA} = 10.2\,\text{mS}$.

## 6.2 SUPPLEMENTARY FIGURES

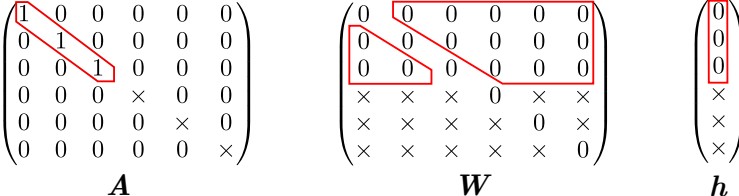

Figure S1: Illustration of the 'manifold-attractor-regularization' for the PLRNN's auto-regression matrix $A$, coupling matrix $W$, and bias terms $h$. Regularized values are indicated in red, crosses mark arbitrary values (all other values set to $0$ as indicated).

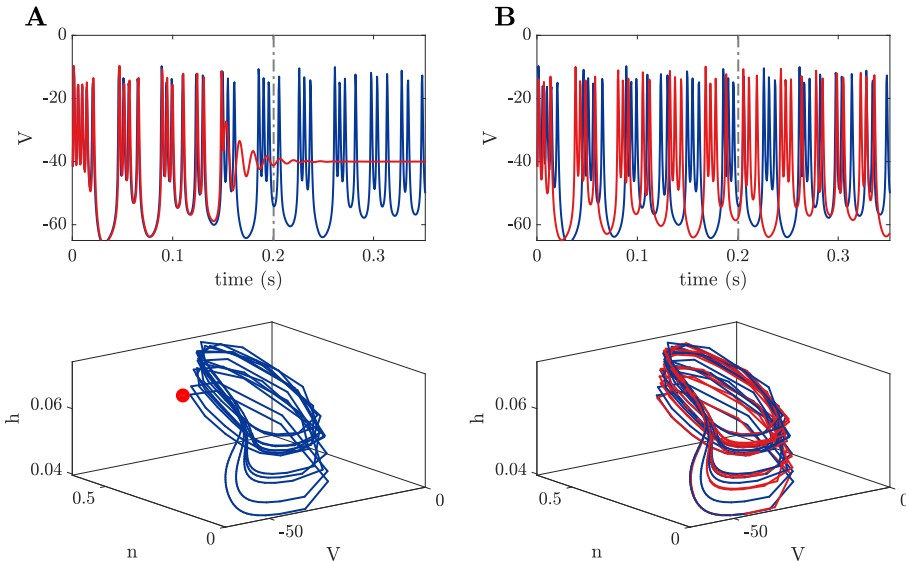

Figure S2: MSE evaluated between time series is not a good measure for DS reconstruction. **A**) Time graph (top) and state space (bottom) for the single neuron model (see section 4.2 and Suppl. 6.1.8) with parameters in the chaotic regime (blue curves) and with simple fixed point dynamics in the limit (red line). Although the system has vastly different limiting behaviors (attractor geometries) in these two cases, as visualized in the state space, the agreement in time series initially seems to indicate a perfect fit. **B**) Same as in **A**) for two trajectories drawn from exactly the same DS (i.e., same parameters) with slightly different initial conditions. Despite identical dynamics, the trajectories immediately diverge, resulting in a high MSE. Dash-dotted grey lines in top graphs indicate the point from which onward the state space trajectories were depicted.

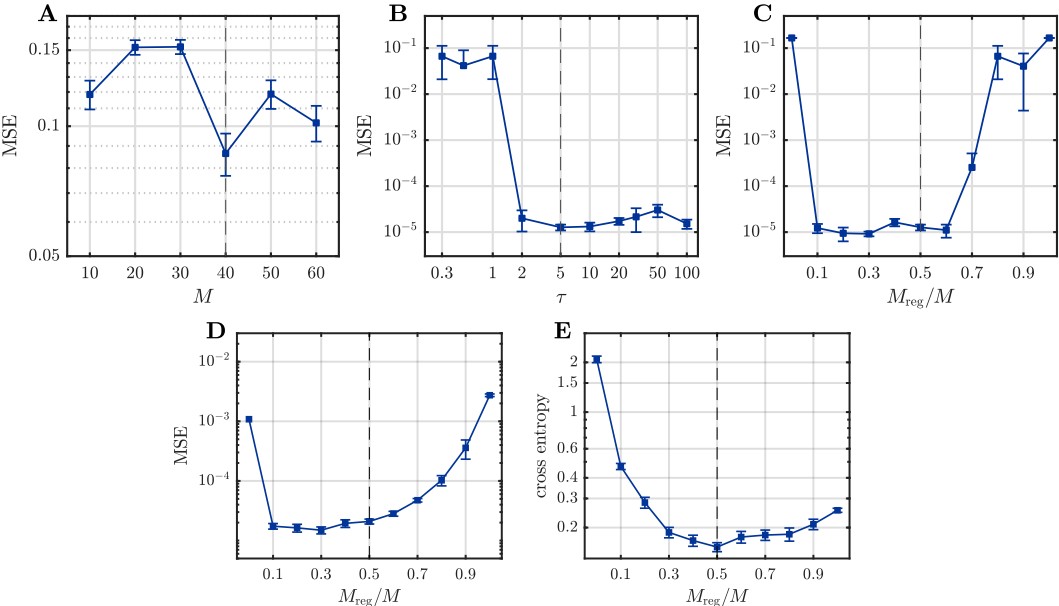

Figure S3: Performance of the rPLRNN for different **A**) numbers of latent states $M$, **B**) values of $\tau$, and **C**–**E**) proportions $M_{\mathrm{reg}}/M$ of regularized states. **A**–**C** are for the addition problem, **D** for the multiplication problem, and **E** for sequential MNIST. Dashed lines denote the values used for the results reported in section 4.1.

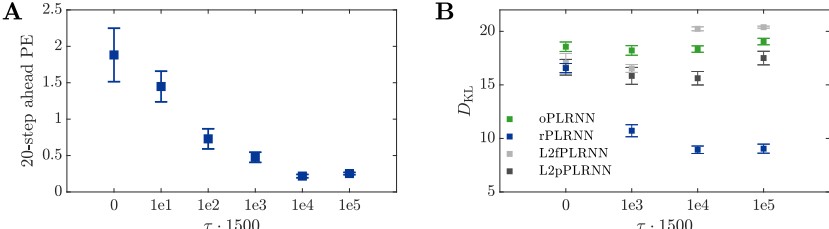

Figure S4: **A**) 20-step-ahead prediction error between true and generated observations for rPLRNN as a function of regularization $\tau$. **B**) KL divergence ($D_{\mathrm{KL}}$) between true and generated state space distributions for orthogonal PLRNN (oPLRNN; i.e., the PLRNN with the 'manifold attractor regularization' replaced by an orthogonality regularization, $(\boldsymbol{A} + \boldsymbol{W})(\boldsymbol{A} + \boldsymbol{W})^T \rightarrow \mathbf{I}$), as well as for the partially (L2p) and fully (L2f) standard L2-regularized PLRNNs (i.e., with all weight parameters for all (L2f) or only a fraction $M_{\mathrm{reg}}/M$ of states (L2p) driven to 0). Note that the quality of the DS reconstruction does not significantly depend on the strength of regularization $\tau$, or becomes even slightly worse, for the oPLRNN, L2pPLRNN and L2fPLRNN. Globally diverging estimates were removed.

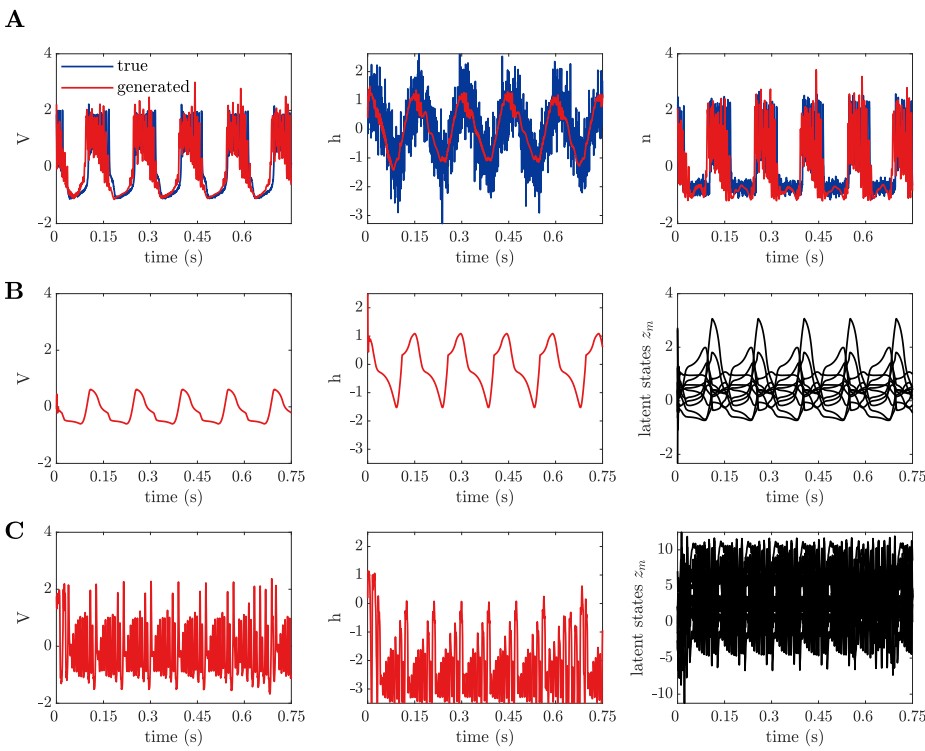

Figure S5: **A**) Reconstruction of fast gating variable $n$ (rightmost) not shown in Fig. 3**D**. For completeness and comparison, other variables have been re-plotted from Fig. 3**D** as well. **B**) Example of reconstruction of voltage ($V$, left) and slow gating ($h$, center) observations, and underlying latent state dynamics (right) for oPLRNN (with orthogonality regularization on $\boldsymbol{A} + \boldsymbol{W}$, see Fig. S4 legend). **C**) Example of $V$ (left) and $h$ (center) observations for standard PLRNN, and underlying latent state dynamics (right). In general, both the standard and the oPLRNN tended to produce many fixed point solutions. In those cases where this was not the case, the standard PLRNN tended to reproduce only the fast components of the dynamics as in the example in **C** (in agreement with the results in Figs. 3**C** & 3**E**), while the oPLRNN tended to capture only the slow components as in the example in **B** (as expected from the fact that the orthogonality constraint tends to produce solutions similar to those obtained for the regularized states only, cf. Fig. 3**E**).

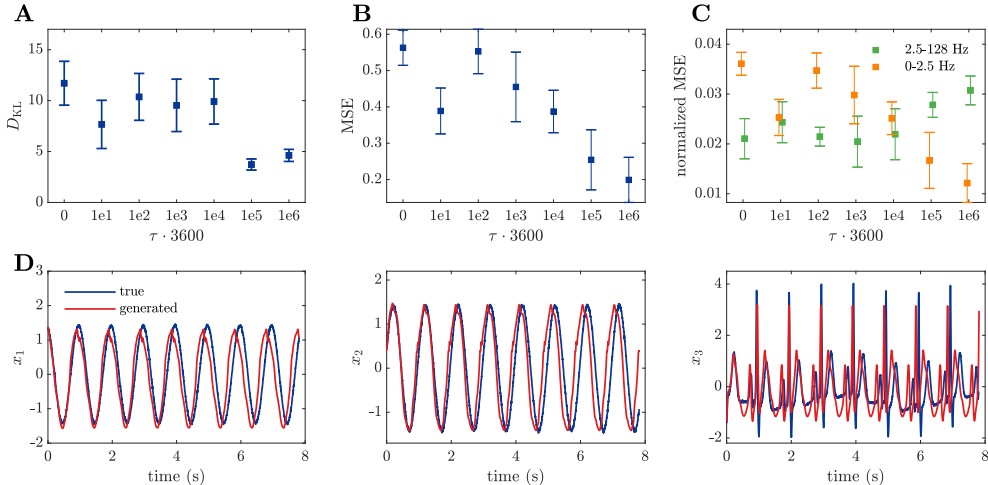

Figure S6: Reconstruction of a DS with multiple time scales like fast spikes and slow T-waves (simulated ECG signal, see McSharry et al. (2003)). **A**) KL divergence ($D_{\mathrm{KL}}$) between true and generated state space distributions as a function of $\tau$. Unstable (globally diverging) system estimates were removed. **B**) Average MSE between power spectra (slightly smoothed) of true and reconstructed DS. **C**) Average normalized MSE between power spectra of true and reconstructed DS split according to low ($\leq 2.5\,\mathrm{Hz}$) and high ($> 2.5\,\mathrm{Hz}$) frequency components. Error bars = SEM in all graphs. **D**) Example of (best) generated time series (standardized, red=reconstruction with $\tau = 1000/3600$).

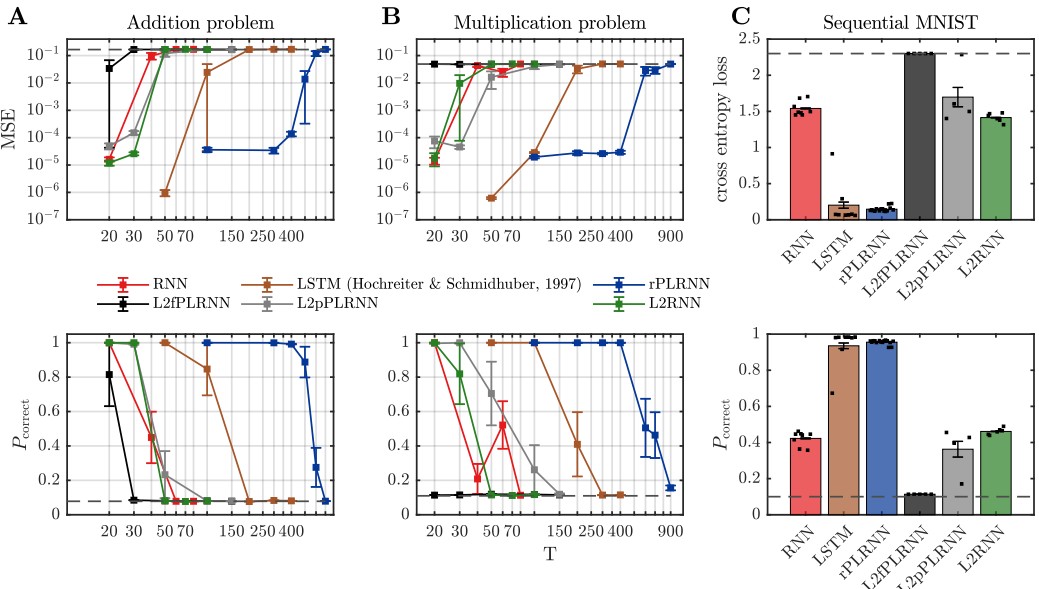

Figure S7: Same as Fig. 2, illustrating performance for L2RNN (vanilla RNN with L2 regularization on all weights) and L2fPLRNN (PLRNN with L2 regularization on all weights) on the three problems shown in Fig. 2. Note that the L2fPLRNN is essentially not able to learn any of the tasks, likely because a conventional L2 norm drives the PLRNN parameters *away* from a manifold attractor configuration (as supported by Fig. 4 and Fig. S8). Results for rPLRNN, vanilla RNN, L2pPLRNN, and LSTM have been re-plotted from Fig. 2 for comparison.

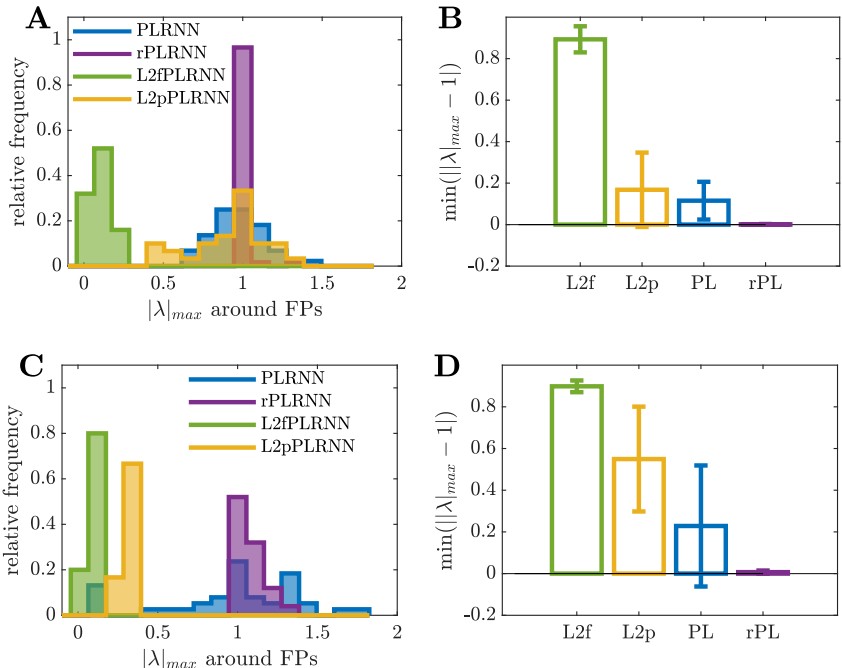

Figure S8: Same as Fig. 4 for **A, B**) multiplication problem, and **C, D**) sequential MNIST. Error bars = stdv.

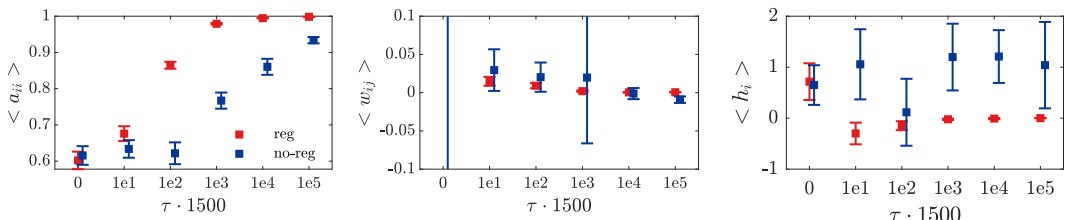

Figure S9: Effect of regularization strength $\tau$ on rPLRNN network parameters (cf. eq. 1) (regularized parameters for states $m \le M_{\text{reg}}$, eq. 1, in red). Note that some of the non-regularized network parameters (in blue) appear to systematically change as well as $\tau$ is varied.

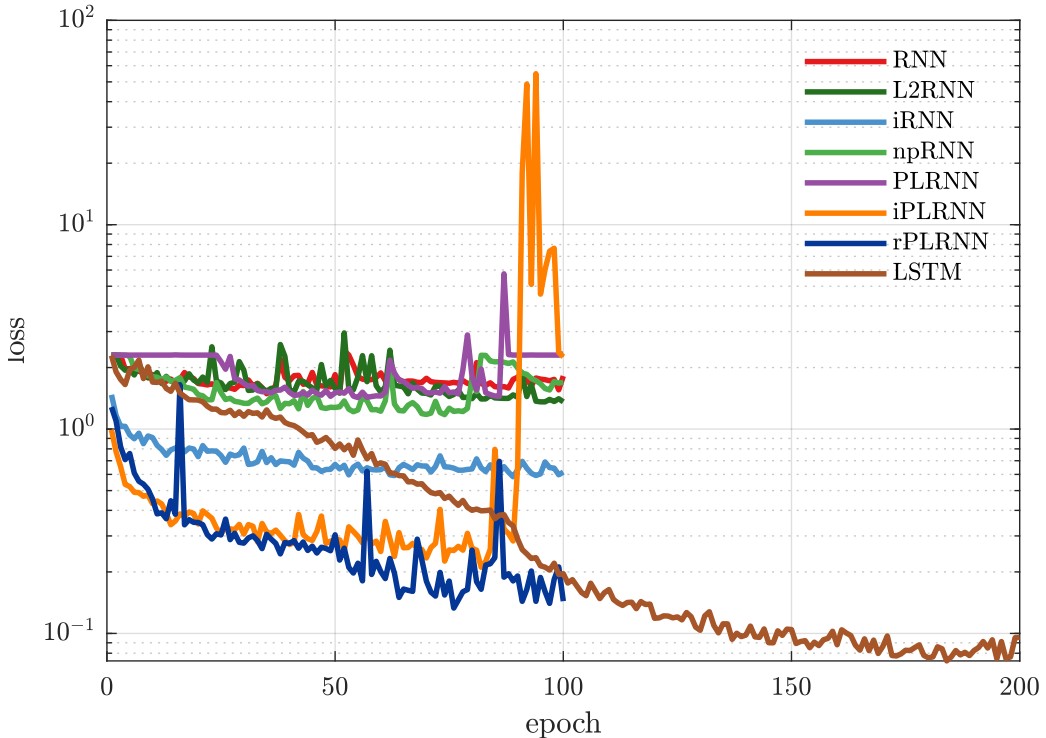

Figure S10: Cross-entropy loss as a function of training epochs for the best model fits on the sequential MNIST task. Note that LSTM takes longer to converge than the other models. LSTM training was therefore allowed to proceed for 200 epochs, after which convergence was usually reached, while training for all other models was stopped after 100 epochs. Also note that although for the best test performance on seq. MNIST shown here LSTM slightly supersedes rPLRNN, on average rPLRNN performed better than LSTM (as shown in Fig. 2**C**), despite having much fewer trainable parameters (when LSTM was given about the same number of parameters as rPLRNN, i.e. $M/4$, its performance fell behind even more).

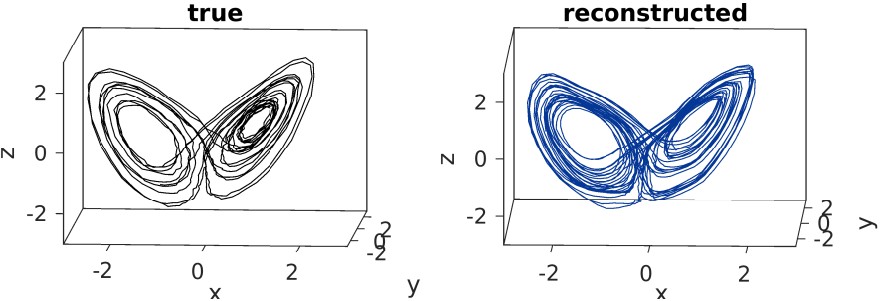

Figure S11: Example reconstruction of a chaotic system, the famous 3d Lorenz equations, by the rPLRNN. Left: True state space trajectory of Lorenz system; right: trajectory simulated by rPLRNN ($\tau = 100/T$, $M = 14$) after training on time series of length $T = 1000$ from the Lorenz system.

