# OpenReview forum: "Identifying nonlinear dynamical systems with multiple time scales and long-range dependencies"
_ICLR.cc/2021/Conference — ICLR 2021 Spotlight_

### Official Review · AnonReviewer3 · 2020-10-27
**Weak Recommendation to Accept**

**Rating:** 7
**Confidence:** 4

**Review:**

This paper proposes a regularization scheme for training vanilla Relu RNN to tackle the exploding and vanishing gradients issue. The work eases the analysis of RNN in the dynamical system point of view and connects the RNN dynamics and gradient theoretically. The experiments show the competitive performance comparing to LSTM.

The vanilla RNNs simplify the analysis as dynamical systems without those gates in LSTMs and GRUs while suffering exploding or vanishing gradients. The idea of tackling the issue  while keeping the simplicity sounds very interesting and useful.

I am leaning on accepting this manuscript if the authors could address my concerns.

- Sec 3.2. The authors mentioned the connection between the particular setting of RNN and working memory. This setting leads to a system without any autonomous dynamics that is not a typical model for working memory (e.g. attracting fixed-points, line attrators and etc.). I disagree that the space is neurally stable since all the states are sensitive to perturbation and cannot persist a stable memory.

- Sec 3.2. Following the so-called "neurally stable" setting, why the term on A in Eq.3 only regularizes the diagonal rather than penalizes the deviation from the identity matrix? This regularization does not lead to the tendency of A -> I mentioned in the text.
Furthurmore, the regularization does not guarantee either A -> I or W -> 0 so that the resulting Mreg subspace does not have the "memory" property.

- Eq.3. How is Mreg determined?

Minor:

- iPLRNN is used before defined.

---

> ### Author Response · Authors · 2020-11-18
> **Clarification added on autonomous dynamics, attractor and memory properties/ properties of matrices, and choice of M_reg**
>
> We thank the referee for the generally positive assessment and appreciation of our work!
>
> 1) *Autonomous dynamics/ stability*: Thanks for bringing this up. However, there seems to be a misunderstanding here: In the absence, or after withdrawal, of external inputs $s_t$ (eq. 1), the system indeed *autonomously* evolves toward line or plane attractor attractor configurations. Please note that the regularization is *only in place during model training* (it is a term added to the loss function used for optimization). After training on data or specific tasks (i.e., after parameter estimation has been completed), the system can be run completely autonomously (as in fact it was in Fig. 3!), or inputs can be provided (like stimuli in real working memory tasks), depending on task context.
> Please also note that we called the regularized configuration *neutrally* stable (not *neurally* stable; as the referee may be aware of, neutral stability is just another term used in the dynamical systems literature (e.g. Strogatz 2015) for marginal stability). To avoid this potential misunderstanding, we have now replaced ‘neutral’ by ‘marginal’ in the revised version. Apart from this, the referee is right in so far as *directly on the line/plane attractor* the system is sensitive to noise perturbations, which is a general property of any line/plane attractor configuration due to its marginal stability. Actually, this has even been observed experimentally in real neural systems (see references in footnote 2)! However, overall the system is still an attractor, as states off from the attractor plane will converge to it (which is why it’s still a model for parametric working memory, e.g. Machens et al. 2005). We made these points explicit now in sect. 3.2 (see footnote) on p. 4.
>
> 2) *Regularization of A and W*: Probably this was not clear enough from our text: $\mathbf{A}$ is a *diagonal* matrix, while $\mathbf{W}$ is *off-diagonal* (see sect. 3.1, below eq. 1, now highlighted in revision). Hence, regularizing only diagonal elements of $\mathbf{A}$ toward 1 will give us $\mathbf{A} \rightarrow \mathbf{I}$, while regularizing the off-diagonal elements of $\mathbf{W}$ toward 0 will give us $\mathbf{W} \rightarrow \mathbf{0}$, as also empirically verified in Fig. S9. We tried to emphasize this again in sect. 3.2 (directly below eq. 3) – does this clarify the referee’s point?
> Furthermore, new Fig. 4 (and new Fig. S8) now verifies empirically that the rPLRNN indeed develops plane attractor configurations (as indicated by the distributions of maximum absolute eigenvalues around fixed points), hence a memory subsystem.
>
> 3) *Choice of M_reg*: Very good point: In general, like is the case for most other meta-parameters of neural models, we would determine it by grid search and generalization error (as in fact we did for our examples, and as was now clarified in the revised version on p. 9, new sect. 4.3). However, as it turns out, there is a larger range of M_reg/M ratios that yields about equally good performance (Fig. S3 C, new Fig. S3 D-E). That is, model performance is not very sensitive to the precise setting of this parameter, and hence for most practical purposes using a value of Mreg/M=0.5 could be recommended. In the revision we elaborated on this point on p. 9, new sect. 4.3.
>
> *Minor*: Thanks for pointing out – we now defined in sect. 4.1 (p. 8) all abbreviations in the main text before first usage.
>
> Finally, we noticed that this referee expressed a ‘weak recommendation to accept’ in her/his summary title (which would correspond to a rating of 6), but the rating given (5) is actually marginally *below* acceptance threshold. We wondered whether this was perhaps by accident?

---

> > ### Comment · AnonReviewer3 · 2020-11-24
> > **My concerns are addressed well**
> >
> > Thanks for the clarification. My concerns are addressed. I would like to recommend acceptance.
> >
> > 2. My point was the regularization only pushes A and W toward I and 0 instead of exact values so that it does not  necessarily satisfy your marginally stable example at the beginning of the paragraph.
> >
> > The title and rating was intentional as I said I would recommend acceptance if my concerns were addressed. I hope it would not discourage the authors.

---

> > > ### Author Response · Authors · 2020-11-25
> > > **Thanks for the supportive reply, we are happy we could address your concerns!**
> > >
> > > Many thanks for your kind response, we are happy we could address your concerns!
> > >
> > > Re pt. 2, this is true, we see what you mean, we started off with a kind of ‘idealized’ situation, which however will indeed be met in the limit $\tau \rightarrow \infty$, and as new Fig. 4 (esp. D) confirms can be met almost arbitrarily closely in practice.
> > >
> > > (… and please don’t forget to update your score :) )

---

> > > > ### Author Response · Authors · 2020-11-25
> > > > **P.S.**
> > > >
> > > > … we added this as a clarifying remark at the end of sect. 3.1.

---

### Official Review · AnonReviewer2 · 2020-10-28
**an interesting novel regularization term that encourages long short-term memory for RNN**

**Rating:** 6
**Confidence:** 4

**Review:**

The paper proposes a novel regularization term to PLRNN. PLRNN has nice numerical properties given its simple mathematical structure, but is able to capture complicate dynamics. It's also easy to establish a theoretical connection between PLRNN dynamics and the behavior of its gradients, which is nice. Given such a dynamic model, they design novel L2 terms pushing partial parameters towards zero, which leads to a line/plane attractor that allows slow time constants for long short-term memory. I think the idea is quite novel and interesting.

A few concerns:
1. It might be a bit jumpy for readers outside neuroscience to establish the connection between working memory and short-term memory. I would say making this point more explicit in the intro would be helpful.
2. Lack of definition of G at the beginning of 3.1.
3. It would be helpful to point out that eq 1 is PLRNN (with full name spell spelled out).
4. Section 4.1 is a bit confusing to me. It's unclear that why the higher the mse/cross entropy is, the better is the model should be; also the lower the Pcorrect value is, the better the model is. Moreover, I think it's still necessary to give the definition of all the model names included in the main text. Some are missing, e.g. iPLRNN, oRNN, maybe just very briefly. Just expect that not all readers would read the appendix.
5. The paper seems to care more about "interpretable", which is not clearly reflected in the paper. Figure 3 only shows the reconstruction. But it would be more interesting to visualize the latents of this neuron model's dynamic. It's mentioned that M={8,...,18} states were trained. How do they look like? Some are more line-attractors and some are more related to the fast spiking dynamics? Which state number is finally picked? Figure 3 only shows how influential \tao is. but there should be other insightful ablation studies to be done in order to understand the regularization term as well, i.e. how many latent states, how to split the two types, etc.
6. It would also be helpful to visualize the reconstruction and the latents for other RNN/LSTM models for the neuron model as well. That helps to show the advantage of the proposed model more clearly.

In sum, I think the idea is quite interesting and practically useful. I also appreciate the theoretical analysis. But the experiment section is confusing by missing some explanations. And the presentation of the neuron model is not sufficient enough to prove that rPLRNN find interesting and interpretable dynamics. That's why I think it's a bit below a good paper.

---

> ### Author Response · Authors · 2020-11-18
> **Clarifications added in revision, latent state dynamics and other dynamical analyses shown in set of new figures (Fig. 3E, 4, S5, S8, new sect. 4.3), choice of M_reg clarified**
>
> Thank you very much for the generally encouraging and supportive comments!
>
> We will address the specific concerns in turn:
>
> 1. Thanks for pointing out. We added a footnote on this (to not interrupt the flow of the main text too much) in the 3rd pg. (p. 2) of the Introduction. We hope this clarifies the connection.
>
> 2. Thanks, now edited in the revised version (in fact, G can be omitted at this point as it is introduced further below).
>
> 3. Now done, see revised version.
>
> 4. Regarding the first point, there is apparently a misunderstanding (we checked our text for typos in this regard). As the referee correctly points out, a *lower* MSE or cross-entropy corresponds to *better* model performance; therefore, the more the curves in Figs. 2A,B are shifted to the right, the better is the performance (as they have lower MSE across a larger range of T). On the other hand, *higher* Pcorrect values (probability of correct responses) are indicative of *better* performance (see definition in sect. 3.5). To make all this clearer, we added a sentence to the Fig. 2 legend which makes these points explicit (see revised version).
> As suggested, we now also spelled out other model abbreviations (iPLRNN etc.) in the main text (sect. 4.1).
>
> 5. Regarding the first point (latent dynamics), we are happy about this suggestion which indeed helped to reveal more about how the regularization works, and how rPLRNN reconstructions are based on the dynamics of the latent states. As requested, we now provide concrete examples of latent state trajectories in new Fig. 3E, discussed in a new pg. at the bottom of sect. 4.2. As evident from this graph, some of the latent states indeed encode very slow time scales, while others are more associated with the fast (spiking) dynamics.
> We would like to stress, however, that what we mean by interpretability is the mathematical tractability and (analytical) accessibility of the rPLRNN to rigorous dynamical systems analysis (see sect. 3.3 and suppl. sect. 6.1.2). This feature is now directly exploited in new Fig. 4 (new sect. 4.3) where we used the theoretical results from sect. 3.3/ suppl. sect. 6.1.2 to derive fixed points and the eigenvalue spectra of their Jacobians analytically, demonstrating that the system forms precise plane attractors with strong regularization (further material on this in new Fig. S8).
> Regarding the second point (choice of M_reg), we now clarified in new sect. 4.3 (p. 9) how to choose M and, in particular, the ratio M_reg/M of regularized to non-regularized states: While one usually would determine these meta-parameters by grid search and cross-validation/ generalization performance, as we demonstrate in Fig. S3C (and in new panels Fig. S3 D-E), there is in fact quite a broad range, roughly $M_{reg}/M \in [0.3, 0.6]$, which works about equally well. Hence, there is not that much of meta-parameter tuning/ searching required, but one would most commonly be on the safe side when choosing $M_{reg}/M=0.5$ for instance. On the other hand, for the extremes M_reg/M-->0 (no regularized states) and  M_reg/M-->1 (all states regularized, no freely evolving states), performance breaks down (see Fig. S3 C-E).
>
> 6. We did evaluate performance of orthogonal RNN and standard PLRNN on this problem as well, perhaps that was not clear enough from our writing (see new Fig. S4 B). However, as requested, we now also show examples of reconstructed observations and latent states for the standard PLRNN and the oPLRNN in Fig. S5. As this figure exemplifies (and as further confirmed with summary statistics in new Fig. S4 B), the plain PLRNN (without regularization) and the oPLRNN essentially were not able to learn this very challenging problem, since they were unable to express either the slow (standard PLRNN) or fast (oPLRNN) components, respectively, in their latent dynamics.
>
> In summary, we have added illustrations of the latent state dynamics (new Fig. 3E, new Fig. S5) as well as eigenvalue spectra around fixed points (new Fig. 4, new Fig. S8) to reveal more about the latent dynamics, have extended the discussion of the bursting neuron model in sect. 4.2, and have added a new sect. 4.3 to deal with other points related to dynamical systems analysis of the effects of the regularization and the choice of M_reg/M (incl. new Fig. S3 D-E).

---

### Official Review · AnonReviewer1 · 2020-10-28
**Interesting paper, but I'm not convinced that the proposed motivation (encouraging plane attractors) is the reason for the improved performance.**

**Rating:** 7
**Confidence:** 5

**Review:**

This paper proposes a type of regularization for piecewise linear RNNs (PLRNNs) that encourages the network to learn line or plane attractors. The paper argues, through mathematical analysis of the regularized network as well as numerical experiments, that this regularization alleviates the vanishing and exploding gradient problem and allows PLRNNs to reconstruct nonlinear dynamical systems with multiple timescales from noisy observations.

Major concern:
I found the paper interesting. My main concern has to do with the motivation of learning line or plane attractors. The paper argues that the proposed regularizer will improve performance *through a specific mechanism: encouraging plane attractors in the dynamics*. While the results in the experiments section are impressive, the paper as far as I can tell does not establish that the regularized PLRNNs have learned plane attractors.

For example, it could be that simply adding l2 regularization on all of the weights would also lead to better performance (rather than the specific regularization proposed), or perhaps the better performance arises from some other (undiscovered) mechanism. In fact, to show that the specific form of l2 regularization is what is useful, I think having a PLRNN with standard l2 regularization (applied to all of the weights) is a critical missing baseline.

To convince me that the benefit is really due to the given motivation (encouraging plane attractors), I want to see evidence that regularized PLRNNs have learned plane (or line) attractors, compared to unregularized PLRNNs (or better yet, PLRNNs with l2 regularization applied to all of the weights). This can be demonstrated in a number of ways, for example by showing eigenvalues of the Jacobian around fixed points of trained PLRNNs with and without the proposed regularization, on the three tasks in Fig. 2. If this was conclusively demonstrated, I would happily increase my rating.

For comparison, recent work showed that RNNs of multiple types (including vanilla RNNs and LSTMs) learned line attractors when solving an NLP task [1]. These line attractors were found in the networks after training, and did not have any special regularization encouraging them. I would appreciate if the authors would add some discussion comparing their regularization to these discovered line attractors.

[1] Maheswaranathan et al, NeurIPS 2019 (http://papers.nips.cc/paper/9700-reverse-engineering-recurrent-networks-for-sentiment-classification-reveals-line-attractor-dynamics)

Other concerns:
- How is Mreg (the number of regularized dimensions) chosen? How does the performance vary as a function of Mreg?
- What's g in eq(2)?
- I was a little surprised that the vanilla RNN worked so well in Fig 2A. I would have expected an LSTM to work as well, if not better. Do the authors have intuition for why this is?

Minor typos:
- Quotes around automatize in first and last paragraphs of section 1 are incorrect. should be `automatize' and `classical'. in latex, use the backtick character (`) for the first quote.
- The abbreviation "RNN" is used as if it is plural, but it is more commonly singular. (e.g. RNNs seem like instead of RNN seem like; or RNNs in their vanilla form instead of RNN in their vanilla form)

---

> ### Author Response · Authors · 2020-11-18
> **Standard L2 baselines added for all tasks (included in previous or new figures), plane attractors now explicitly verified through examination of eigenvalue spectra (new Figs. 4 & S8, new sect. 4.3), broader discussion on plane attractors added**
>
> We thank the referee for the very helpful and constructive comments!
>
> 1) *L2 regularization/ missing baseline*: This is a fair point, and we have now added this baseline to Fig. 2 of the paper, adding further results on this in a new Fig. S7 (and also for the bursting neuron DS in new Fig. S4 B). As the new results now presented in Fig. 2 and Fig. S7 demonstrate, a PLRNN with standard L2 regularization performs much worse than our rPLRNN and most of the other models. This is irrespective of whether we place a standard L2 norm on all weights (Fig. S7) or just on a proportion of M_reg/M states (updated Fig. 2) as for our own regularization (eq. 3).
> The bad performance of the PL/RNN with conventional L2 regularization on *all* weights is particularly interesting and revealing: It may be rooted in the fact that driving all weights to 0 actually pushes the RNN *away* from a plane attractor configuration, as supported by the results in new Figs. 4 & S8 (see reply to point 2 below).
>
> 2) *Demonstrating plane attractors*: As suggested by the referee, we now quantified for all problems in Fig. 2 and Fig. 3 how close our proposed rPLRNN compared to standard and L2-regularized PLRNNs is to a plane attractor constellation by examining the distribution of absolute eigenvalues from all fixed points (computed exactly using eq. 8 in theoretical suppl. sect. 6.1.2). We added this as a new Fig. 4 to the main text (with additional results on other tasks in new Fig. S8), discussed in new sect. 4.3. As can be seen, the rPLRNN has a large proportion of maximum absolute eigenvalues centered on 1 (confirming plane attractor configurations) which increases with the regularization parameter $\tau$. In contrast, absolute eigenvalues of plain or L2-regularized PLRNNs are much more broadly distributed. Perhaps even more important, Fig. 4 B&D (as well as Fig. S8) furthermore confirm that also the plane attractor tuning is much more precise with larger regularization, with the eigenvalues deviating much less from 1 as regularization increases, and as compared to plain and L2-regularized PLRNN. Finally, Fig. S9 also demonstrates that parameters in the rPLRNN are indeed pushed toward a configuration that would support a plane attractor subspace as the regularization increases (while Fig. 4 might make this point more explicit).
>
> 3) *Discussion on plane attractors in other RNNs*: Thanks for pointing this out, we have now included this work in our Conclusions sect. (p. 9). Yes, other networks including plain RNNs or PLRNNs can certainly develop line/plane attractors to solve specific tasks, we observed this as well. However, plain PL/RNNs are much harder to train to achieve this (at least on the tasks we tested), are usually successful only on a smaller percentage of runs (due to the very nature of the vanishing/exploding gradient problem), and – perhaps most importantly in the context of long-range dependencies – exhibit much less precise plane attractor tuning than our rPLRNN (i.e., with max. abs. eigenvalues deviating further from 1, cf. Fig. 4 B&D, Fig. S8 B&D). Gating-based architectures like LSTMs, on the other hand, lack the mathematical simplicity and tractability of the PLRNN, which was another major motivation for our work.
>
> Other concerns:
>
> - *Choice of M_reg*: Good point – see our reply to the second point of referee #4: As most meta-parameters in R/NN models, one could determine the optimal settings by grid search and testing generalization performance (as we did). However, as Fig. S3C demonstrates, there is actually quite a broad range of M_reg/M across which performance is nearly optimal (in essence, one should mainly avoid the extremes). We now confirmed this also for other of the presented experiments, as illustrated in new panels Fig. S3 D-E. Hence, choosing an intermediate value of $M_{reg}/M=0.5$ should most commonly be fine. This is now discussed more extensively in new sect. 4.3.
>
> - *g in eq.2*: Thanks, now clarified.
>
> - *RNN performing well in Fig. 2A*: This must be a misunderstanding. Perhaps there was a confusion of the different line colors, or we didn’t make our abbreviations (specifically iRNN) clear enough: Vanilla RNN is indeed the *worst* performing model (red line), LSTM is in fact among the better performing models. iRNN, in contrast, stands for identity-initialized RNN, which was already shown to be better than LSTMs on this task by Le et al. (2015). We spelled out more of these abbreviations now in the main text and hope this helps to avoid this misunderstanding (Fig. 2 legend was also expanded).
>
> Many thanks for pointing out the typos, all corrected now in the revision.

---

> > ### Comment · AnonReviewer1 · 2020-11-25
> > **thanks for your response!**
> >
> > Thank you for your response, as well as the additional experiments. They address most of my initial concerns. I have updated my score to reflect this.

---

> > > ### Author Response · Authors · 2020-11-25
> > > **Thanks for your kind response and support!**
> > >
> > > We are happy to hear we have covered most of your concerns – thanks again for your time and your help in improving this manuscript!

---

### Official Review · AnonReviewer4 · 2020-10-30
**Excellent paper proposing a novel regularization technique to ReLU-based RNNs for dynamical system identification**

**Rating:** 8
**Confidence:** 4

**Review:**

The paper explores a very important question in dynamical system identification of how to make recurrent neural networks (RNNs) learn both long-term and short-term dependencies without the gradient vanishing or exploding limitation. They suggest using piece-wise linear RNNs (PLRNNs) with a novel regularization technique.

The paper is well written and is very thorough with the necessary theoretical foundation, numerical experiments and analysis.

I think the theory and results of this paper are significant and will be relevant to further our understanding of RNNs and system identification.

Major points:

1) L2 weight regularization can be easily applied to any of the RNN models used in the experiments. While other weight initialization schemes were compared to the paper's proposed model (rPLRNN), none of the other RNN models had similar regularization. This will shed some light on whether it is indeed the proposed regularization that matters or the full proposed model with PLRNN and a mix of regularized and non-regularized units.

2) It is not clear to me how one can choose the correct ratio of regularized vs unregularized units in the model. While the amount of regularization clearly helps in reducing training error as shown in Figure 3, increasing the ratio of regularized units in Figure S3C did not help the error past 0.1 and then larger values resulted in large increases such that the error at ratio 1 is equivalent to the error at ratio 0. Perhaps this observation is specific to the addition problem, but I feel that a discussion of the effect of this ratio on performance should be included for clarity. Additionally, the ratio of regularized units with best performance could potentially be different for different regularization amounts.

Minor Point:
g is not defined in equation 2

---

> ### Author Response · Authors · 2020-11-18
> **Specific role of regularization and choice of M_reg/M now discussed with several new results and figures**
>
> We thank the referee for the very supportive and constructive comments!
>
> 1. We included two sets of new results in the revised manuscript that should address the first point. First, we now checked that a standard L2 regularization (pressing all weights to 0) won’t do the job: As updated Fig. 2 and new Figs. S4 B & S7 demonstrate, standard L2 regularization by itself (L2RNN, L2fPLRNN) leads to performance about as bad as for vanilla RNN. Second, if we place a standard L2 regularization just on a subset of M_reg states of the PLRNN (L2pPLRNN; as we did with our proposed regularization, eq. 3), performance is again down to (or worse than) that of the plain PLRNN (updated Fig. 2, Fig. S4B). Hence, the specific form of the regularization eq. 3 is indeed important. New Figs. 4 and S8 furthermore reveal more about the mechanisms by which the regularization worked.
> On the other hand, only for this particular form of model (PLRNN) the specific regularization eq. 3 ensures that a subspace of the latent space is pushed toward a manifold attractor configuration (cf. new Fig. 4). As we had briefly noted in passing in sect. 3.2 (p. 5), the particular form of the PLRNN may be advantageous to begin with (supported by observations reported in sect. 4.2). For some of the other models a similar regularization as eq. 3 could potentially be designed, but this is not the case for all of them (e.g., orthogonality constraints on the full matrices would rule out a-priori a subspace in which the model can freely evolve, and hence phenomena like chaos). Also, at least for LSTMs it would give up on one other key advantage of our formulation, namely its analytical and dynamical tractability.
> Finally, it’s also important we have a mix of regularized and non-regularized states: If we had only regularized states (and regularization strongly enforced), the system would have a harder time to learn specific tasks or express certain types of dynamics, as empirically verified in Fig. S3 C-E (new Fig. 3E furthermore demonstrates the ‘division of labor’ among regularized and non-regularized states for the bursting neuron DS).
> We discussed these new results and points in more detail in the revised version in new sect. 4.3 (p. 9) and in the Conclusions (sect. 5).
>
> 2. Yes, thanks, very good point. As suggested, this is now discussed in more detail, with new results, in new sect. 4.3 (p. 9). Briefly, like any meta-parameters in other types of models, one would usually determine the optimal settings by grid search and testing generalization performance (as we did in the present case). However, as Fig. S3C demonstrates, there is actually quite a broad range of M_reg/M across which performance is nearly optimal (in essence, one should mainly avoid the extremes of M_reg/M-->0 or M_reg/M-->1 as the referee correctly noted). We now confirmed this tendency also for other of the presented experiments, as illustrated in the new panels Fig. S3 D-E. Hence, one mainly needs to ensure one has an about equal mix of both regularized and non-regularized states in the model, but the specific ratio is not so important and any intermediate value $M_{reg}/M \in [0.3, 0.6]$ should most commonly be fine.
>
> Minor Point: Thanks for pointing out, now fixed!

---

### Author Response · Authors · 2020-11-18
**General reply to referee comments**

We thank all four referees for the careful reading of our manuscript, and for all the valuable feedback and suggestions! We have thoroughly revised the manuscript according to the comments: We added a total of 4 new results figures and 4 new subpanels/ results to previous figures, a new results section (4.3), as well as several new paragraphs and additional explanations elsewhere in the text. To make room for all this new material, in addition to using p. 9, we had to shorten the previous Conclusions (and some other sections) a bit. For clarity, we also restructured and expanded the proofs of the theorems a bit.
We are thankful for the many suggestions which helped to further strengthen a number of points in the paper and to reveal in more depth how our regularization approach works.

---

> ### Author Response · Authors · 2020-11-24
> **Latest Revision**
>
> Update: Few more clarifications added, minor typos fixed

---

### Decision · Program_Chairs · 2021-01-07
**Final Decision**

**Decision:**

Accept (Spotlight)

**Comment:**

This paper describes a clever new class of piecewise-linear RNNs that contains a long-time scale memory subsystem. The reviewers found the paper interesting and valuable, and I agree. The four submitted reviews were unanimous in their vote to accept. The theoretical insights and empirical results are impactful and would be suitable for spotlight presentation.